# Specific and nondisruptive interaction of guanidium-functionalized gold nanoparticles with neutral phospholipid bilayers

Lucía Morillas-Becerril[1,6], Sebastian Franco-Ulloa [2,5,6], Ilaria Fortunati[1], Roberto Marotta[3], Xiaohuan Sun[4], Giordano Zanoni [1], Marco De Vivo [2✉] & Fabrizio Mancin [1✉]

Understanding and controlling the interaction between nanoparticles and biological entities is fundamental to the development of nanomedicine applications. In particular, the possibility to realize nanoparticles capable of directly targeting neutral lipid membranes would be advantageous to numerous applications aiming at delivering nanoparticles and their cargos into cells and biological vesicles. Here, we use experimental and computational methodologies to analyze the interaction between liposomes and gold nanoparticles (AuNPs) featuring cationic headgroups in their protecting monolayer. We find that in contrast to nanoparticles decorated with other positively charged headgroups, guanidinium-coated AuNPs can bind to neutral phosphatidylcholine liposomes, inducing nondisruptive membrane permeabilization. Atomistic molecular simulations reveal that this ability is due to the multivalent H-bonding interaction between the phosphate residues of the liposome's phospholipids and the guanidinium groups. Our results demonstrate that the peculiar properties of arginine magic, an effect responsible for the membranotropic properties of some naturally occurring peptides, are also displayed by guanidinium-bearing functionalized AuNPs.

[1] Dipartimento di Scienze Chimiche, Università di Padova, via Marzolo 1, Padova, Italy. [2] Laboratory of Molecular Modeling and Drug Discovery, Istituto Italiano di Tecnologia, Via Morego 30, Genoa, Italy. [3] Electron Microscopy Facility (EMF), Istituto Italiano di Tecnologia, Via Morego 30, Genoa, Italy. [4] School of Chemistry and Chemical Engineering, Yangzhou University, Yangzhou, Jiangsu, People's Republic of China. [5]Present address: Expert Analytics. Møllergata 8, Oslo, Norway. [6]These authors contributed equally: Lucía Morillas-Becerril, Sebastián Franco-Ulloa. ✉email: marco.devivo@iit.it; fabrizio.mancin@unipd.it

The ability of nanoparticles to interact with the surfaces of cells, microorganisms, and viruses plays a key role in their biological activity. Cationic nanoparticles generally show a higher affinity for lipid membranes and better internalization rates[1–4]. These properties are usually ascribed to the electrostatic attraction between the nanoparticles and the negatively charged membranes[5–10].

In this context, properties of small (<5 nm) gold nanoparticles (AuNPs) coated with a shell of cationic ligands (usually trialkyl ammonium headgroups) are particularly interesting. These AuNPs were indeed reported to be easily taken up by cells[11]. Studies with model membranes suggested that they can induce disruption of the lipid bilayer, which might also explain the observed cytotoxicity[12–18].

The peculiarity of ligand shell protected AuNPs extends beyond their net charge, as evidenced by the behavior of their anionic counterpart. Indeed, Stellacci and others showed in the last decade that also alkyl-sulfonate coated AuNPs can penetrate cell membranes, embed within synthetic bilayers, or induce hemifusion between the vesicles, depending on the particle size and ligand shell conformation[5–9].

The message emerging from these examples is that the entire chemical structure of the nanoparticle's coating ligands, i.e., both the headgroups and the inner alkyl chains, plays a role in the interaction with lipid bilayers.

Interestingly, properties similar to those of cationic nanoparticles are shared by other polycationic entities, such as antimicrobial peptides, cell-penetrating peptides (CPPs), polymers, and dendrimers[11,19–21]. These macromolecules can bind to cells and spontaneously pass through the plasma/endosomal membranes. To do so, they use different mechanisms that range from pore formation, as with most antimicrobial peptides[22], to transient bilayer disruption, as with several CPPs[23]. Also, CPPs may benefit from the presence of hydrophobic adjuvants, as pyrenebutyrate[24–26]. The chemical structure of the positively charged groups plays a critical role in the molecule's effectiveness in translocating across membranes. The best internalization rates are usually achieved when guanidinium is the charge-bearing group, so that the ability of arginine, the amino acid containing this group as a side chain, to increase the cell penetration capacities of macromolecules was named "arginine magic"[27,28]. The origins of this effect are still debated. Likely, it is the result of several concurrent properties of guanidinium. First, its $pK_a$ value in water is 13.6, ensuring full protonation in almost all the conditions that can be encountered in a biological environment. Second, the structure of guanidinium provides two parallel oriented H-bonds donors, which are almost perfectly pre-organized for a cooperative interaction with oxyanions as carbonate, phosphate, and sulfate. In these cases, the H-bonding interaction is further reinforced by charge polarization and ion-pairing interaction. Third, the planar structure of this group leads to an unusual solvation, with the water molecules lying on the molecular plane and the faces acquiring a hydrophobic character. Finally, guanidinium has the unusual ability to form contact pairs in water, despite the charge repulsion. Taken together, these features ensure that guanidinium-rich molecules have a high positive charge in most conditions (deprotonation induced by charge proximity is prevented by the high $pK_a$), can strongly interact with the phosphate groups present in the phospholipids forming biological membranes, and can alter the bilayer structure thanks to its peculiar solvation and self-aggregation properties.

The conceptual similarity between cell-penetrating polymers and cationic AuNPs was recognized early on[17]. Indeed, one should expect that also guanidinium-functionalized AuNPs should have peculiar properties when interacting with biological membranes. To date, however, AuNPs coated with guanidinium groups have only been investigated as artificial phosphodiesterases and macromolecular receptors[29–32].

Here, we report the results of a comparative study revealing that small AuNPs coated with guanidinium ligands can interact nondisruptively with neutral phospholipid bilayers. Our experiments demonstrate that electrostatic interactions are not the main factor affecting the binding of nanoparticles to membranes. Using experiments and molecular dynamics (MD) simulations, we show that the distinctive features of guanidinium nanoparticles are due to the specific ability of this functional group to recognize the phosphate group of the lipids at the membrane.

## Results

**AuNP functionalization and characterization.** The main goal of this study was to evaluate the interaction between lipid bilayers and cationic nanoparticles and to elucidate the structure−activity relationship at play. We, therefore, selected a small library of cationic AuNPs coated with ligands 1−7 (Fig. 1). The ligands featured alkyl linkers of different lengths and either guanidinium (1−3) or trimethylammonium (4−7) headgroups. Synthesis of 1−7 (Supplementary Methods, Section 1.2, and Supplementary Schemes 1−7) and 1−7@AuNPs followed standard procedures (Supplementary Information, Section 1.3, and Supplementary

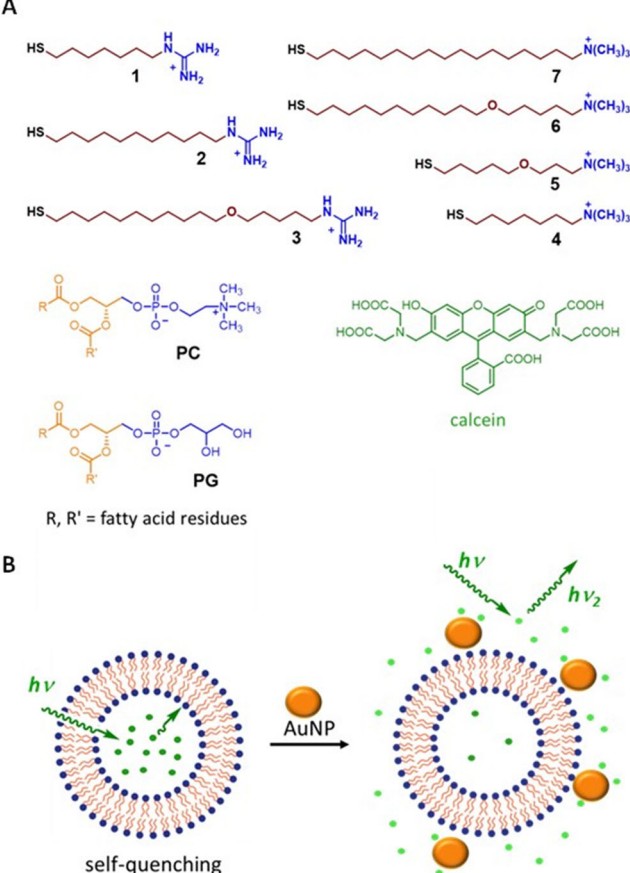

**Fig. 1 Chemical species and processes investigated in this study.**
**A** Structures of the nanoparticle-coating thiols 1−7, of the calcein dye, and of the liposome-forming phospholipids used in this study. **B** Schematic representation of the calcein release experiments: emission from calcein molecules confined in the liposomes is reduced by concentration quenching, the interaction of nanoparticles with the liposomes induces the escape of part of calcein from the inner water pool, with a consequent dilution of the dye and increase of the emission.

| Table 1 Characterization data for 1−7@AuNP[a]. | | | | | |
|---|---|---|---|---|---|
| AuNPs | Core diameter (nm) | Z-potential (mV)[b] | Ligands per NP | Gold atoms per NP | Ligands footprint (nm²)[c] |
| 1@AuNP | 2.4 ± 0.6 | 11.6 ± 3.2 | 260 ± 13 | 427 ± 64 | 0.07 ± 0.04 |
| 2@AuNP | 1.9 ± 0.4 | 9.2 ± 0.6 | 165 ± 8 | 212 ± 32 | 0.07 ± 0.03 |
| 3@AuNP | 2.4 ± 0.8 | 9.1 ± 1.3 | 73 ± 4 | 427 ± 64 | 0.25 ± 0.17 |
| 4@AuNP | 1.6 ± 0.3 | 8.1 ± 0.7 | 66 ± 3 | 127 ± 19 | 0.12 ± 0.05 |
| 5@AuNP | 1.6 ± 0.3 | 6.4 ± 2.2 | 44 ± 2 | 127 ± 19 | 0.18 ± 0.07 |
| 6@AuNP | 1.5 ± 0.3 | 11.4 ± 0.9 | 53 ± 4 | 104 ± 16 | 0.13 ± 0.03 |
| 7@AuNP | 2.1 ± 0.6 | 10.7 ± 0.5 | 106 ± 6 | 286 ± 43 | 0.13 ± 0.08 |

[a]See part 3 of Supplementary Information for details on calculations, errors are the standard deviations.
[b]Measured at 25 °C in 10 mM HEPES buffer, pH 7.0, containing 100 mM NaCl.
[c]Average surface area occupied by each ligand.

Figs. 2−18). The size of the gold cores in 1−7@AuNPs ranged in the 1.5−2.4 nm interval, according to TEM measurements (Table 1 and Supplementary Table 1). Thiolate footprints (the average gold surface occupied by a single thiolate) obtained were in the 0.1−0.2 nm² interval, which well compares with typical values of alkylthiol-protected gold nanoparticles. The Z-potential values (in HEPES/NaCl buffer) ranged from +6.4 to +11.6 mV, confirming the cationic coating of the particles (Table 1 and Supplementary Table 1). All the nanoparticles were well soluble in water and in the buffered solutions used in this study. In the rest of the paper, nanoparticle concentrations will be expressed as total concentrations of the coating ligands ([ligand]) to account for the small differences in the size of 1−7@AuNP.

**Liposome permeabilization assays**. To initially evaluate the interaction between neutral liposomes and AuNPs, we used the dye release assay (Fig. 1), which is typically used for this purpose[27,33−39]. In this assay, a fluorescein derivative is entrapped at a high concentration in the inner water pool. Inside the liposome, the emission of the dye is strongly reduced by a concentration-induced self-quenching. If an agent capable of perturbing the lipid bilayer, allowing the escape of the dye from the water pool, is added to the samples, the consequent dilution of the dye in the bulk solution reduces the effectiveness of the self-quenching process and produces an increase of the fluorescence emission. For several polycationic species, the ability to allow dye release correlates with a cell-penetrating ability[27,37,38]. In our experiments, we used liposomes made with phosphatidylcholine (PC, Fig. 1C) and we encapsulated in the water pools calcein, a polyanionic fluorescein derivative (Fig. 1). Liposomes were prepared by extrusion with 100 nm polycarbonate filters, and they displayed an average hydrodynamic diameter of 90 nm and a Z-potential close to zero (−1.7 mV), as expected for neutral lipids like PC (Supplementary Table 2).

We performed preliminary fluorescence experiments by incubating calcein-loaded liposomes with 1@AuNP. These experiments showed a clear increase in emission, revealing the particle's ability to induce dye release. The final emission values observed in each experiment increased with the concentration of AuNPs to eventually level off at [ligands] around 30−40 µM. Since the concentration of PC in the samples is 22 µM, this corresponds to a [ligands]/[PC] ratio of ca. 2 (Supplementary Fig. 20). The calcein release process was very fast, concluding within 1 or 2 min after the AuNP addition (Supplementary Fig. 21). Two different batches of 1@AuNP with different average sizes (i.e., 2.4 ± 0.6 and 1.4 ± 0.2 nm) were tested providing similar results (Supplementary Fig. 22).

Based on these results, we set the [ligands]/[PC] ratio to 3 for the subsequent experiments and we systematically investigated the membrane perturbation ability of 1−7@AuNPs (Fig. 2, red).

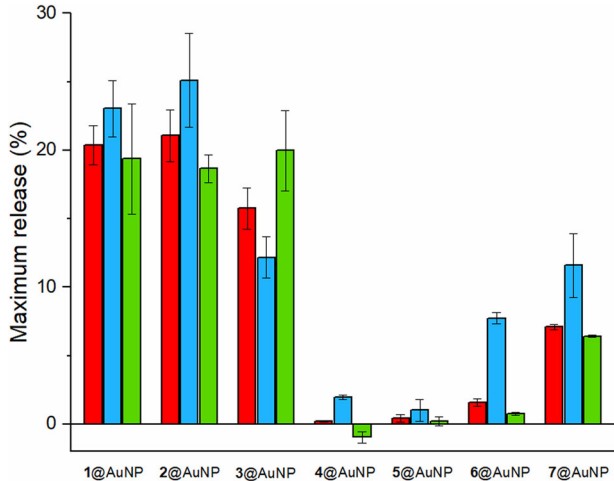

**Fig. 2 Calcein release from liposomes.** Maximum release of calcein from PC (red), PC/PG (blue), and PC/Chol (green) liposomes after addition of 1−7@AuNPs. Experimental conditions: [HEPES] = 10 mM, [NaCl] = 100 mM, [PC]/ [PC + PG]/ [PC + chol] = 22 µM, [AuNPs-thiols] = 66 µM, pH 7.0, 25°. Error bars represent standard deviations calculated from triplicate experiments.

Remarkably, the results clearly show that guanidinium-bearing AuNPs always led to the escape of a larger amount of calcein than their trimethylammonium counterparts. Specifically, 1, 2, and 3@AuNP-induced emission recoveries close to 20% (with respect to the emission increase obtained by disassembling the liposomes with the Triton-X100 surfactant), whereas the emission increase was negligible (less than 2%) for 4, 5, and 6@AuNP, and only slightly greater (about 7%) for 7@AuNP.

Since all the nanoparticles had a similar charge, these results reveal that the interaction between cationic AuNPs and neutral lipid bilayers is governed by factors other than the coating molecule's charge. Namely, the AuNP's membrane perturbation activity is modulated by variations in the chemical structure, including the nature of the cationic headgroups and the features of the underlying chains.

**Liposome's structural integrity**. To get more information on the nature of the permeabilization process, we investigated the structural integrity of the liposomes in the presence of the nanoparticles. First, to ensure that the effects observed were not due to leaky or unstable liposomes, we investigated the effect of adding cholesterol (Chol) to the bilayers. Chol is present in a relevant amount (about 25−30%) in mammalian cell membranes and helps to increase their stability and barrier ability[40,41]. We repeated the release experiments using PC/Chol 10:3 liposomes (97 nm diameter, −0.7 mV Z-potential). The

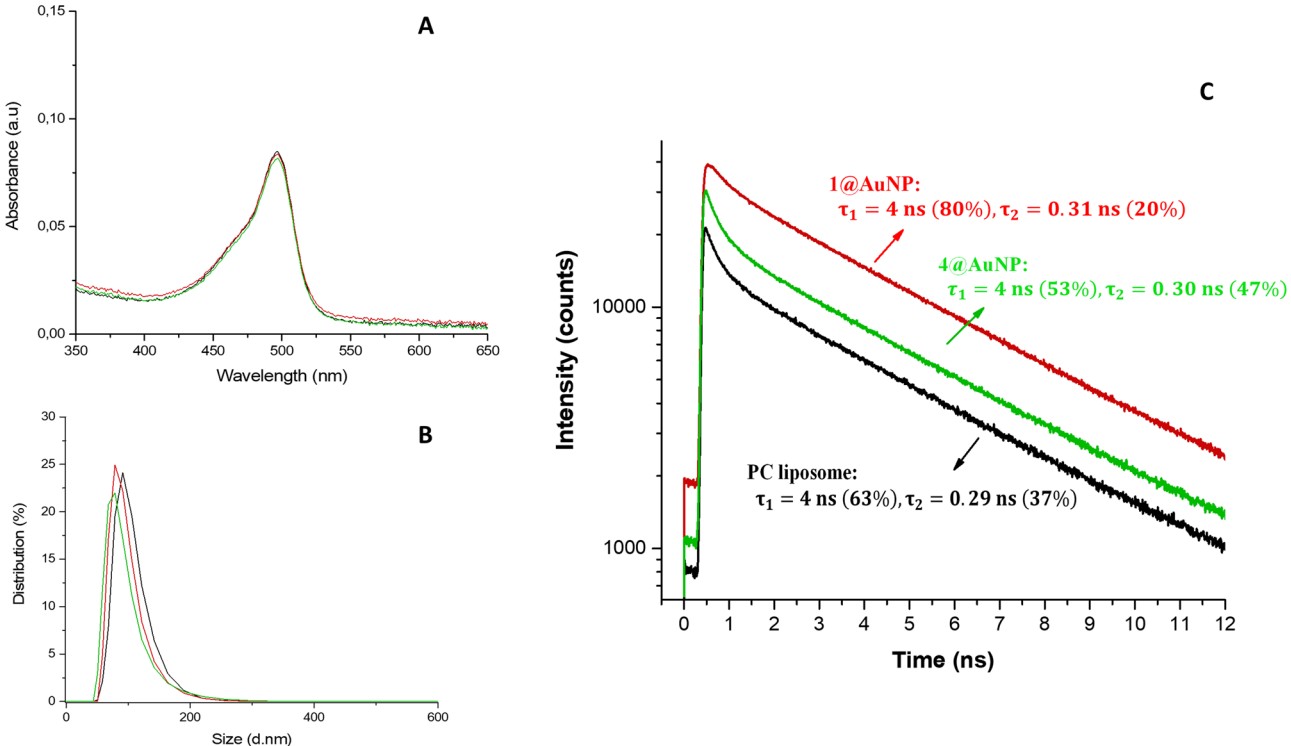

**Fig. 3 Liposomal integrity. A** UV−Vis spectra of samples containing calcein-loaded PC liposome samples (black), after the addition of 4@AuNPs (green), and after the addition of 1@AuNPs (red). **B** DLS size distribution diagrams of samples containing calcein-loaded PC liposome samples (black), after the addition of 4@AuNPs (green), and after the addition of 1@AuNPs (red). **C** Fluorescence decay curves of calcein-loaded PC liposomes alone (black), after the addition of 4@AuNPs (green), and after the addition of 1@AuNPs (red) under the same excitation power. The relative amplitude of each component is in parentheses. Experimental conditions: [HEPES] = 10 mM, [NaCl] = 100 mM, [PC] = 22 µM, [AuNPs-thiols] = 66 µM, pH 7.0, 25°C.

results (Fig. 2, green) showed in all the cases emission recoveries similar to those measured in the absence of Chol, revealing that the ability of guanidinium nanoparticles to induce calcein escape persisted even on less fluid bilayers.

Subsequently, we analyzed the samples with UV−Vis spectroscopy and dynamic light scattering (DLS). UV−Vis spectra (Fig. 3A) revealed that the addition of nanoparticles did not produce any visible change. In particular, no increase in the baseline intensity, as expected in the case of liposome aggregation, was observed. DLS analyses confirmed this information showing that, in all our samples, the PC liposomes retained a nearly constant hydrodynamic size and dispersion index (PDI) when incubated with 1−7@AuNP at the concentrations used in the calcein release experiments (Fig. 3B and Supplementary Figs. 24, 25). Thus, the DLS measurements endorsed the structural stability of the liposomes upon nanoparticle addition.

As a further control, the effect of increasing the nanoparticle concentration was investigated for 1 and 4@AuNP (Supplementary Figs. 26, 27). Here too, the hydrodynamic size and PDI were unaffected for nanoparticle concentrations (expressed as the total concentration of coating ligands) of up to 1 mM, i.e., 20-fold the concentration used in the calcein release experiments. Only in the case of 1@AuNP, but not of 4@AuNP, when the nanoparticles concentration was raised to 1 mM, hydrodynamic size sensibly increased, suggesting that at these very high concentrations, guanidinium nanoparticles might induce liposomes aggregation or degradation.

Additional information was provided by fluorescence lifetime measurements on the same samples under 440 nm pulsed excitation. In all cases (Fig. 3C), we detected two emission decay processes. The first process had a longer lifetime (4 ns), and was assigned to residual unquenched calcein (indeed, the same

lifetime was measured for a dilute calcein solution). The second process had a shorter lifetime (0.3 ns) and was assigned to the self-quenched dye molecules entrapped within liposomes.

The identification of the short-living species as-quenched entrapped dyes was confirmed by measuring the lifetimes of the most intense emission spikes recorded during intensity trace measurements (Supplementary Fig. 28 and Supplementary Table 3). These spikes corresponded to events, where several liposomes were simultaneously illuminated by the laser beam, leading to a steep increase in the light emitted. As such, they provided information on the state of the entrapped dyes. During the spikes, the lifetimes composition substantially changed with the fraction of the short-living species, substantially increasing and reaching up to 90%.

Upon addition of AuNPs, the ratio of lifetime populations in liposome samples underwent no significant modifications with 4@AuNP (Fig. 3C, green trace), confirming that liposomes are unaffected by the presence of these nanoparticles. However, the addition of 1@AuNP resulted in a clear increase in the fraction of unquenched dyes, as expected in the case of a release of entrapped calcein (Fig. 3C, red trace).

We also performed confocal microscopy experiments (Supplementary Fig. 29). Samples showed a dotted emission arising from the diluted liposomes. This pattern was unchanged after the addition of 1@NP and 4@AuNP, but with 1@AuNP a stronger background emission, due to the released dye, was observed.

Taken together, fluorescence emission experiments, DLS measurements, and confocal microscopy point to a permeabilization mechanism, where the interaction of the nanoparticles with the liposomes induces a local destabilization of the double layer. This causes the release of the entrapped dye molecules without affecting the overall structural integrity of the liposomes.

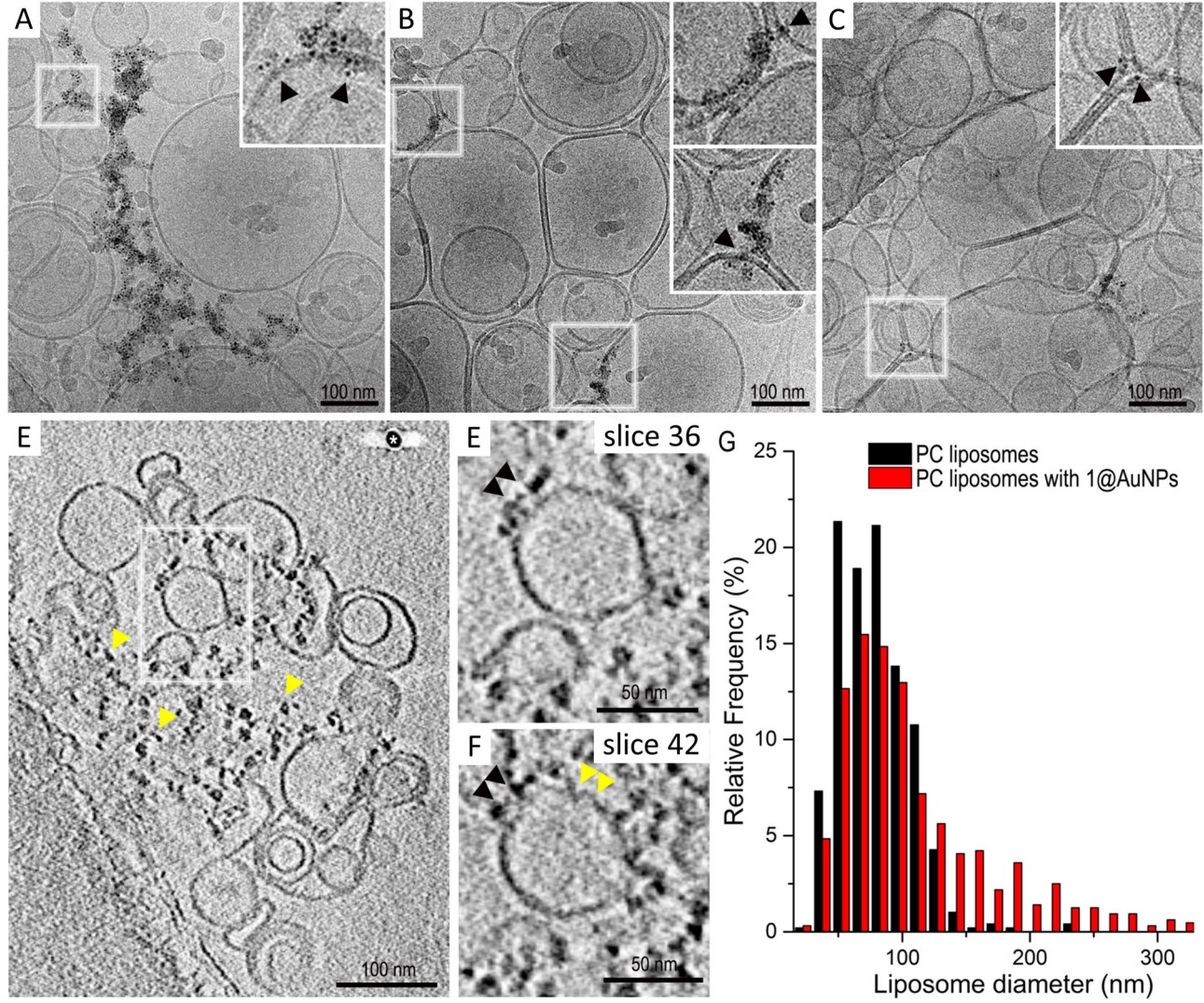

**Fig. 4 Cryo-EM and cryo-electron tomography analysis of PC liposomes with 1@AuNPs. A−C** Cryo-EM projection images showing the presence of NPs associated with the liposomes. The insets are higher magnifications of the corresponding boxed regions. Note the presence of NPs interacting with the liposome membranes (black arrowheads); **D** cryo-electron tomography averaged image (ten adjacent slices) of liposomes interacting with several NPs (yellow arrowheads). The asterisk points to a 15 nm AuNP used as a fiducial marker (see also Supplementary Video 1 in SI); **E, F** are high-magnification images (average of ten adjacent slices) of the boxed region in (**D**) at different levels inside the tomogram. Note in (**E**) and (**F**), respectively, the presence of NPs interacting with the liposome membrane (black arrowheads) and the liposome membrane partially perturbed (yellow arrowheads); **G** size distribution of unloaded PC liposomes (black) and PC liposomes incubated with 1@AuNPs (red).

**The nature of the liposome−nanoparticle interaction**. To shed light on the supramolecular structure of the nanoparticle−liposome complex, we investigated our samples with cryo-electron microscopy (cryo-EM, Fig. 4). Samples were first vitrified in liquid ethane, to preserve their structural integrity. They were then analyzed with transmission electron microscopy at cryogenic temperature (i.e., below −170 °C). Micrographs of PC liposomes revealed the presence of subspherical liposomes in all samples. The average diameter of the PC liposomes was 79 ± 27 nm ($n = 492$), in agreement with the dimensions measured with DLS experiments (Fig. 4G and Supplementary Fig. 19). In addition to the expected unilamellar vesicles, we observed the presence of more complex structures, such as nested multilamellar vesicles. Such structures were present in all the samples and therefore could not be due to the presence of the nanoparticles.

When the cryo-EM analysis was performed on samples containing PC liposomes and AuNPs, we found that the addition of nanoparticles did not significantly modify the shape (Fig. 4A−E) and size of the liposomes (Fig. 4G), but for the formation of a small fraction of larger liposomes. Interestingly, for 1@AuNPs but not for 4@AuNPs, we detected regions where small clusters of 2 nm NPs lay near the liposomal membrane (Fig. 4A−C and their respective insets). To avoid the ambiguity of projection images, we performed cryo-electron tomography to investigate whether these pictures showed an interaction between 1@AuNPs and liposomal membranes (Fig. 4D−F and Supplementary Video 1). This analysis confirmed the presence of NPs mainly adsorbed on the external liposome membrane (Fig. 4D−F). In addition, the regions of the membrane that were interacting with the AuNPs were partially broken or perturbed (Fig. 4F). Such perturbations could be at the origin of the dye release and the formation of larger liposomes.

Cryo-EM results thus confirmed the preferential adhesion of 1@AuNP to liposomes, suggesting a greater affinity of these nanoparticles for PC membranes.

**The role of electrostatic attraction**. Based on these results, we decided to inspect the role of electrostatics in the nanoparticle's

ability to interact with liposomes. We repeated the calcein release experiments with liposomes comprising negatively charged phosphatidylglycerol (PG, Fig. 1) and PC in a 1:9 ratio. These liposomes had an average hydrodynamic diameter of 100 nm and, as expected, a negative Z-potential of −7.7 mV (in PBS buffer, pH 7.0). Introducing a net negative charge into the liposomes would, in principle, enhance the electrostatic attraction to cationic nanoparticles. Indeed, for cell-penetrating peptides, the presence of PG in the liposomes has been reported to increase their interaction with the bilayer and thus their double-layer perturbation activity[3,20,22,23]. Interestingly, release experiments (Fig. 2, blue bars) revealed negligible differences between PC and PC/PG liposomes for guanidinium 1, 2, and 3@AuNPs. However, the long-chain ammonium-bearing nanoparticles 6 and 7@AuNP induced a greater calcein release from PG-containing liposomes, when compared to the pure PC vesicles. Finally, very small effects were observed for short-chain nanoparticles 4 and 5@AuNP. The secondary relevance of charge attraction in the interaction of neutral liposomes with guanidinium nanoparticles was confirmed by release experiments performed with liposomes suspended in salt-free solutions (we used glucose to equilibrate the osmotic pressure). We found (Supplementary Fig. 23) that calcein release from PC liposomes induced by 1 and 4@AuNP was similar in the presence of glucose or NaCl. On the other hand, in the case of PC/PG liposomes, the release observed in the absence of NaCl was somewhat larger.

The interaction of AuNPs with liposomes was further investigated by measuring the fluorescence emission of PC and PC/PG liposomes loaded with nile red (Supplementary Fig. 1), a hydrophobic dye that locates within the bilayer and cannot be released. AuNPs can effectively quench the emission of dyes that are sufficiently close (within 1−3 nm) to the gold core. The binding of the AuNPs to the liposomes should therefore decrease the sample emission[18]. The results closely paralleled those of the calcein release experiment. Indeed, the addition of 1@AuNP to nile red-loaded PC liposomes (Supplementary Fig. 30, red bars) resulted in a 30% quenching of emissions, while 4@AuNP produced only a 13% quenching, and 6@AuNP and 7@AuNP produced a 20% decrease. For PC/PG liposomes (Supplementary Fig. 30, blue bars), the net charge present on the liposomes had no effect on 1 and 4@AuNPs, which continued to produce relevant and marginal quenching of emission, respectively. However, quenching clearly increased for 6 and 7@AuNP, reaching the level observed for 1@AuNP. Hence, this experiment indicated that the guanidinium nanoparticles bound to neutral and negatively charged liposomes, the long-chain ammonium NPs bound to negatively charged liposomes only, and the short-chain ammonium NPs did not bind to any liposome. Interestingly, this trend closely matches the one observed in calcein release experiments.

To get more insight into the peculiar interaction of guanidinium nanoparticles with neutral PC liposomes, we performed release experiments in the presence of dimethyl phosphate at 10 mM concentration (Supplementary Fig. 31). This anion reproduces the features of the phosphate diester residue present in the PC headgroup. We hence expected it to interact with guanidinium nanoparticles preventing by competition the binding of nanoparticle's guanidinium residues to the lipid's phosphate groups[24]. Indeed, we found that the presence of dimethylphosphate completely leveled the extent of calcein release, removing any difference between guanidinium and ammonium nanoparticles.

This experimental evidence provides key insights into the interactions that govern the nanoparticle−membrane association. Evidently, nanoparticles must bind to the liposomal membrane to produce the permeabilization effects. DLS, confocal microscopy, and cryo-EM experiments confirm that this binding does not affect the liposomes integrity, and likely involves the adhesion of the AuNP to the outer surface. One would therefore expect that positively charged nanoparticles would not interact with neutral liposomes but would bind to negatively charged liposomes. However, we observed that guanidinium nanoparticles bind to and permeabilize neutral liposomes. Moreover, this activity is not enhanced by additional net charge attraction. Ammonium nanoparticles positively interact with negatively charged liposomes only, and only when coated with long-chain thiols. This suggests that guanidinium nanoparticles can bind to neutral bilayers via interactions other than Coulomb pairing. These interactions provide an affinity so great that the additional attractions with PC/PG liposomes do not lead to increased permeabilization activity. Ammonium nanoparticles do not bind to neutral liposomes, but they should bind to PC/PG liposomes. Under our experimental conditions, charge attraction is not sufficient, and the additional presence of long-chain thiols is required. This indicates that effective binding of nanoparticles to liposomes requires stronger interactions, such as those provided by combining headgroup electrostatic attractions with hydrophobic interactions, formed between the ligand alkyl linker and the inner region of the double layer.

**Resolution of atomically detailed pairing between NPs and liposomes**. To investigate the molecular origin of the interactions between our cationic AuNPs and lipid membranes, we performed equilibrium atomistic molecular dynamics (MD) simulations.

Starting from atomistic models[42], we resolved the temporal evolution of the nanoparticle−membrane association to rationalize the exceptional behavior of guanidinium-bearing AuNPs. We considered 1@AuNP, 4@AuNP, and three additional models (4a/4b/4c@AuNP, Fig. 5A) with intermediate capacities for H-bonding. In detail, in the latter three models, we replaced the trimethylammonium headgroup of 4@AuNP with dimethylammonium (4a@AuNP), monomethylammonium (4b@AuNP), and ammonium (4c@AuNP). This allowed us to investigate the properties of protonated amines as cationic nanoparticle headgroups, which are present in CPPs and related species but are experimentally inaccessible in AuNPs. We first performed MD simulations of AuNPs alone in water (100 ns for each of the five

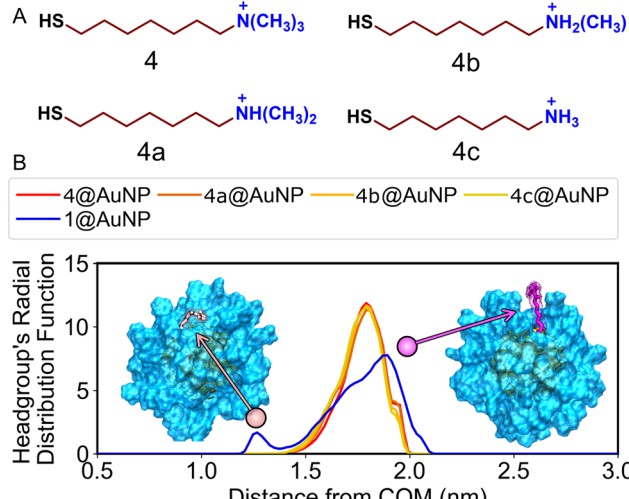

**Fig. 5 Structure of cationic ligand-protected nanoparticles. A** Chemical structure of the thiols forming the monolayer of the simulated AuNPs. **B** Radial distribution function (RDF) of the AuNP's headgroups as a function of the distance from the gold core center of mass (COM). The ammonium derivatives (4/4a/4b/4c@AuNP) display a single peak corresponding to an extended conformation (purple), while the guanidinium-bearing AuNP (1@AuNP) also presents a coiled configuration (pink).

systems) in order to equilibrate their structures and properties. The convergence of the AuNP structure was determined by the stabilization of the root-mean-squared deviation (RMSD) of the atomic coordinates (Supplementary Fig. 33)[43,44]. The cationic headgroups of all the ligands caused a reciprocal repulsion. Indeed, by computing the radial distribution function (RDF) of the head-groups with respect to the gold core's center of mass (COM), we found that the ligand shell of all five AuNPs displayed a predominantly extended conformation with a maximum at c.a. 1.8 nm (Fig. 5B).

However, for 1@AuNP, the headgroup's RDF profile was significantly broader, with a shoulder at 1.6−1.7 nm and a second maximum at 1.3 nm. These differences suggest a coiled arrangement of the thiols. Indeed, some ligands adopted a bent conformation, with the headgroup embedded within the hydrophobic core of the monolayer. On average, this configuration appeared for around 3% of the sampled structures, which corresponds to two ligands (out of 60) per frame. The appearance of this coiled conformation, albeit with low frequency, is likely due to the amphiphilic nature of guanidinium, which acts as a hydrogen bond donor along its molecular plane and as a hydrophobic patch at the plane's faces.[28] At the same time, the presence of folded ligands enhances the amphiphilic nature of 1@AuNPs by inducing the exposition of the inner alkyl linkers to the bulk solution.

Subsequently, equilibrated AuNPs were allowed to freely interact with a 1-palmitoyl-2-oleoyl-sn-glycero-3-phosphocholine (PC) in ~1-µs-long MD simulations for each system. Notably, no external force or potential gradients were applied in these experiments. Remarkably, and in line with the experimental evidence, only 1@AuNP spontaneously associated with the membranes during our simulations (Fig. 6 and Supplementary Fig. 34). In detail, the south pole (here defined as the nanoparticle's region closest to the membrane's surface) of 1@AuNP came into close contact with the membrane's headgroups ~165 ns after the simulation started, and it remained stably bound to the bilayer for the rest of the simulation. In three additional replica simulations, we observed the same phenomenon taking place at 25, 420, and 555 ns. The RMSD of the AuNPs also remained stable during these simulations, endorsing the stability of the models (Supplementary Fig. 35).

Nanoparticle binding to the membrane was always irreversible in the time span of our MD simulations. Simulations of the other AuNPs (i.e., 4@AuNP, 4a@AuNP, 4b@AuNP, and 4c@AuNP) showed no spontaneous binding with the membrane within the sampled timescale, suggesting a less favorable association. Tracking the distance between the AuNPs and the membrane in their respective simulations (Fig. 6) revealed that all AuNPs explored states of similar proximity when they were freely diffusing in the solvent. This confirms that all the nanoparticles can get close enough to the bilayer surface to initiate binding. Nevertheless, the actual anchoring only occurs for 1@AuNP, in our simulations.

The binding of 1@AuNP to the double layer is expected to induce a substantial reorganization of the species that are forming, or strongly interacting with, the AuNPs monolayer, namely the guanidinium headgroups, the chloride counterions, and the water molecules. To get more details on this point, we computed the number density of each component with respect to the AuNP polar angle and quantified its deviation from a perfectly uniform distribution (Fig. 7A and Supplementary Fig. 36). These calculations revealed that the southernmost part of the monolayer (but not the south pole) became more populated with guanidinium groups upon binding. Hence, the ligand shell reorganized to maximize the number of guanidinium −phosphate interactions. Several H-bonds with the lipid

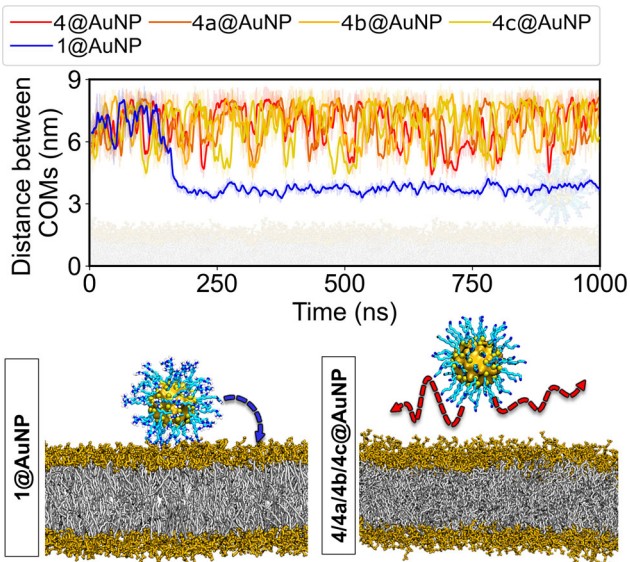

**Fig. 6 Distance between the COM of the membrane and the COM of the gold atoms.** The curve shown for 1@AuNP corresponds to its first replica simulation. The other three replicas display similar results (Supplementary Fig. 34). 1@AuNP is the only particle to spontaneously bind to the PC bilayer even though they all explore states of sufficient proximity to the bilayer. Gold atoms are displayed in orange, sulfur in yellow, carbon in cyan, nitrogen in blue, hydrogen in white, PC headgroups in brown, and hydrophobic lipid tails in gray (see Supplementary Video 2).

headgroups were rapidly formed as soon as 1@AuNP was adsorbed onto the bilayer (Fig. 7B and Supplementary Fig. 37), confirming the establishment of a multivalent interaction. The gathering of guanidinium groups led to a high positive charge density at the south pole of the AuNP. This charge accumulation was stabilized by the chloride counterions that agglomerated in the neighboring subequatorial region. Consequently, the pairing between the ligands and the chloride ions ushered the water molecules toward the north pole of the monolayer. Notably, upon binding, the composition of the monolayer remained unchanged, including the number of embedded chloride ions and water molecules. These results were observed in all four replica simulations with 1@AuNP (Supplementary Fig. 36).

As noted, the contact between 1@AuNP and the lipid bilayer triggered the formation of several H-bonds between the guanidinium-terminated ligands and the phosphate groups of PC (Fig. 7B and Supplementary Fig. 37). We quantified the persistency of such H-bonds when 1@AuNP passed from being fully solvated to being adsorbed on the membrane in each of the replica simulations (Supplementary Table 3). Interestingly, the total number of H-bonds formed by the guanidinium groups when in solution or bound at the membrane was similar ($217 \pm 8$ vs $224 \pm 11$, respectively for the first replica). Hence, H-bonding with membrane headgroups nicely compensated the loss of nanoparticle solvation. Still, there was a subtle repartitioning between the different nanoparticle donors in the two states. In solution, the $\eta 1$ and $\eta 2$ nitrogen atoms of guanidinium formed a total of $177 \pm 7$ H-bonds with water, whereas the $\varepsilon$ nitrogen atom formed $40 \pm 4$ H-bonds (Fig. 7C and Supplementary Table 3). When bound to the membrane, the $\eta$ positions formed $159 \pm 8$ and $23 \pm 5$ H-bonds with the solvent and the phosphate groups, respectively. The $\varepsilon$ position formed $38 \pm 5$ and $4 \pm 2$ H-bonds with water and phosphates, respectively. Therefore, the total number of H-bonds involving the $\eta$ positions slightly increased to $182 \pm 10$ (vs $177 \pm 7$) and that of H-bonds involving the $\varepsilon$ positions slightly increased to $42 \pm 4$ (vs $40 \pm 4$). Even if the total number of

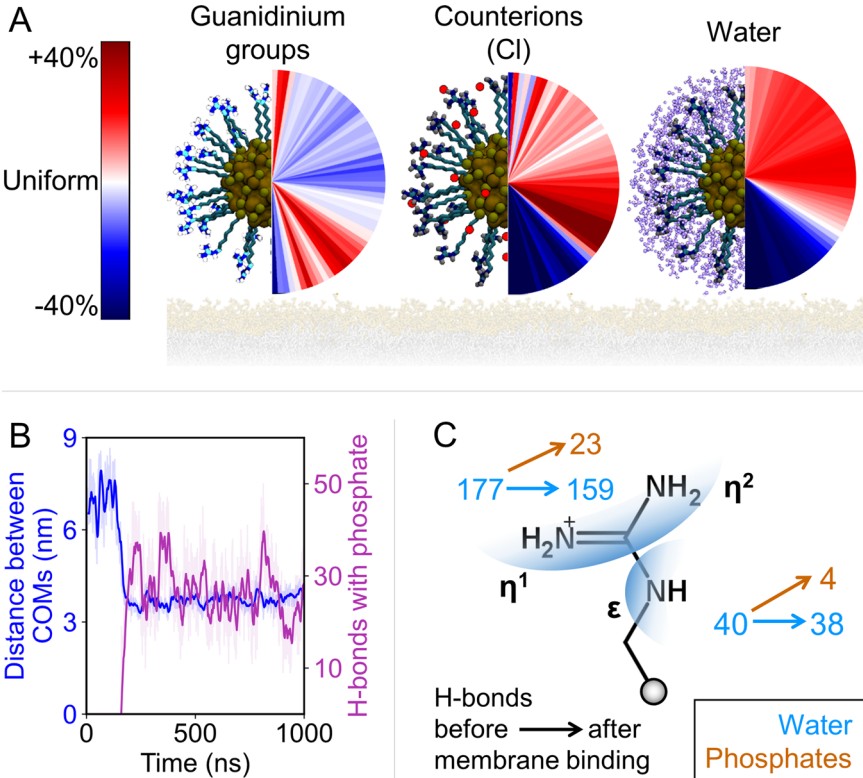

**Fig. 7 Effect of membrane binding on the monolayer of 1@AuNP in the first replica simulation. A** Angular (polar) number density relative to a perfectly homogeneous distribution. In the color bar, white indicates a perfectly uniform distribution in the spherical slice, red is a highly populated region, and blue is the least dense region. Headgroup carbons are shown in cyan, chloride ions in red, and water molecules in purple. **B** Distance between the bilayer's COM and the COM of the gold atoms (blue), as well as the H-bonds between 1@AuNP and the lipid bilayer (violet). The approach of 1@AuNP triggers an H-bond network that stabilizes the bound complex. **C** Change in the number of H-bonds before and after membrane binding for both distinct donor positions in the guanidinium group.

H-bonds remains almost constant, the formation of $27 \pm 7$ bonds between the nanoparticle and the bilayer is likely the driving force of the binding to the membrane, since charge-assisted and cooperative phosphate—guanidinium H-bonds are stronger than solvation interactions of phosphate/guanidinium groups. In line with experiments, the membrane integrity was preserved during all our replica simulations. The membrane thickness (Supplementary Figs. 38,39) and the lipid order parameter (Supplementary Figs. 40, 41) remained within the expected thermal fluctuations. However, upon 1@AuNP binding, the membrane suffered slight distortions. These may foreshadow the enhanced permeation that leads to substantial calcein release at longer timescales.

We could demonstrate these distortions by monitoring two angles, $\alpha$ and $\beta$ when 1@AuNP was adsorbed on the lipid bilayer. The first angle, $\alpha$, measures the orientation of the PC groups with respect to the membrane's normal ($Z$) axis (Fig. 8A), as routinely implemented in such analyses[45]. In our unperturbed membrane, this angle adopted an equilibrium value of 67° (Fig. 8B and Supplementary Fig. 42). When 1@AuNP sat on the bilayer, the $\alpha$ angle increased by around 10°, reaching 77° in all the replica simulations, indicating that the nanoparticle fished the phosphate groups out of the bilayer to optimize the H-bond network (Fig. 8C).

The angle $\beta$ was defined to track the orientation of the PC group with respect to the $XY$ projection of the gold atom's COM (Fig. 8D). In this case, we observed a local decrease of around 30° with respect to the 90° angle expected for a random distribution of an unperturbed membrane (Fig. 8E and Supplementary Fig. 42). The reduction in $\beta$ suggests that the phosphate groups lean inward to the nanoparticle's contact region and that the

ammonium groups point outward (Fig. 8F). Thus, the phosphatidylcholine group of the lipids adopts an arrangement reminiscent of the electrical double layer formed by electrolytic solvents around charged bodies[46]. The affinity between the guanidinium and phosphate groups causes a partial polarization at the contact region, despite the membrane having a neutral net charge. Notably, this effect can only be captured when the AuNPs and the lipids are modeled at an atomistic resolution[17,45–48].

We further investigated the role of electrostatics in nanoparticle—membrane binding by performing four 1-μs-long replica simulations of 1@AuNP freely interacting with a PC/PG (9:1) lipid bilayer. Again, the RMSD of the nanoparticles during these simulations endorsed the stability of our models (Supplementary Fig. 35). In these simulations, 1@AuNP bound spontaneously (and permanently) to the lipid bilayer; this time 44, 45, 57, and 318 ns after the four replica simulations had started (Supplementary Fig. 43). Moreover, 1@AuNP and the membrane produced nearly identical responses to those observed in our simulations with pure PC bilayers. First, the guanidinium headgroups agglomerate on the southern hemisphere of the monolayer, displacing the chlorine ions and water molecules toward the northern regions (Supplementary Fig. 44). Second, the binding of the AuNP onto the membrane is triggered by an H-bond network between guanidinium and phosphate groups (Supplementary Fig. 45 and Supplementary Table 4), while the organization of acyl lipid tails remains unaltered (Supplementary Fig. 46). Third, the headgroups of the PC lipids reorganize, causing the local polarization of the membrane and stabilizing 1@AuNP's charge (Supplementary Fig. 47). Notably, the binding of 1@AuNP to our PC/PG (9:1) bilayer shows a local slimming of the membrane

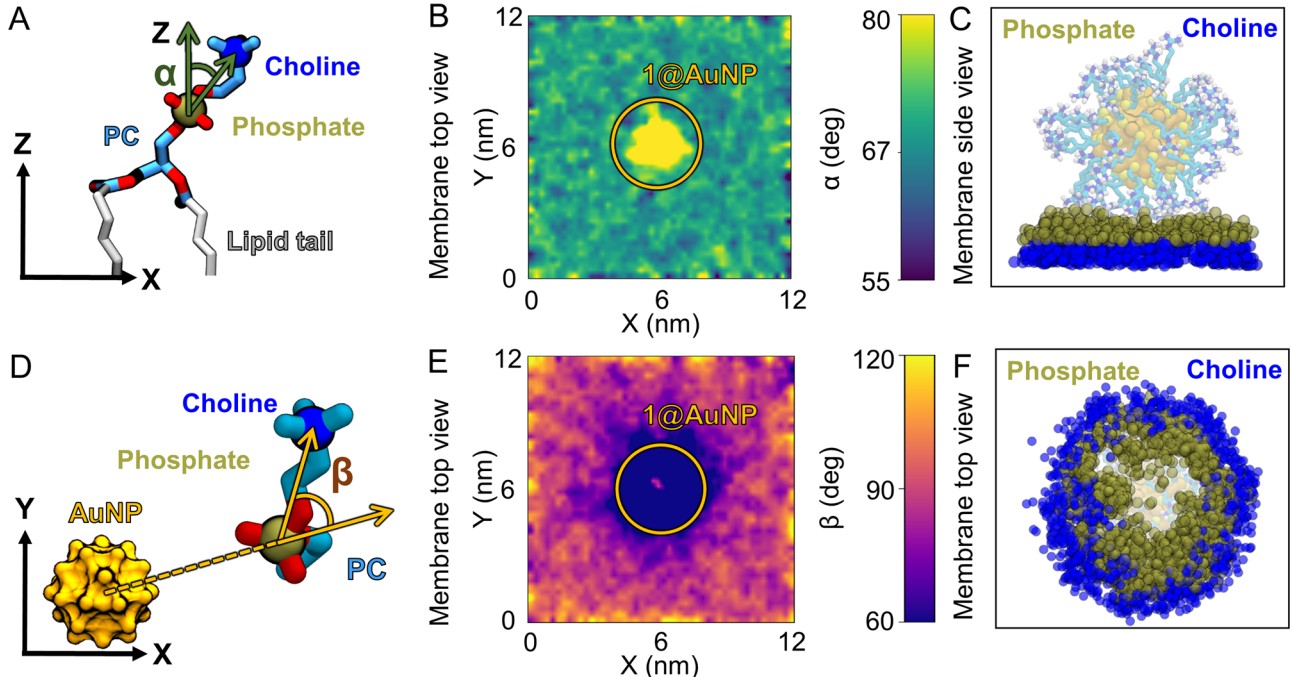

**Fig. 8 Membrane deformations upon binding of 1@AuNP in the first replica simulation. A** Definition of $\alpha$ as the angle between the PC group and the direction normal to the bilayer (Z-axis). **B** Map of $\alpha$ upon nanoparticle binding, as seen from the top (X and Y are the dimensions of the simulation box, as in (**E**)). There is a local decrease of $\alpha$ in the contact region. **C** Conceptual illustration of the phosphate fishing made by superimposing 300 trajectory frames. **D** Definition of $\beta$ as the angle between the XY projection of the vector between the AuNP COM and the phosphate group, and the vector between the phosphorus and nitrogen atoms of the PC group. **E** Map of $\beta$ upon nanoparticle binding from the top. There is a local decrease of $\beta$ around the contact region. **F** Conceptual illustration of the membrane polarization near the nanoparticle binding spot.

underneath (Supplementary Fig. 48). This result suggests that other phenomena, e.g., the translocation of guanidinium-bearing AuNPs, may be promoted in the presence of multicomponent membranes. Taken together, our experimental and computational results endorse the importance of local atomically detailed structural features in AuNP−membrane interactions, which cannot be ascribed to electrostatics only.

## Discussion
The first finding of this study is that the binding of cationic AuNPs to liposomes resulted in their permeabilization. This behavior is explained by MD simulations that suggest that the adhesion of a nanoparticle to the outer surface of a phospholipid bilayer causes local distortions that could trigger the permeabilization.

Second, we showed that AuNPs coated with a monolayer of guanidinium-bearing ligands are highly efficient in interacting with neutral phospholipid bilayers and inducing their non-disruptive permeabilization. This activity is substantially greater than that of other cationic nanoparticles and, notably, occurs irrespective of the bilayer's charge. Hence, guanidinium AuNPs, in contrast to other cationic AuNPs, are capable of binding to neutral bilayers. MD simulations indicate that the driving force could be the ability of these AuNPs to establish a multivalent interaction with the liposome's phosphate moieties. They also suggest that the H-bond donor arrangement of guanidinium, which is perfectly fit for the interaction with phosphate, is crucial in allowing the nanoparticle to bind to the membrane. Hence, as in the case of cell-penetrating peptides and polymers, the "arginine magic" is at play with nanoparticles too, and the guanidinium headgroup has a fundamental role as a phosphate-recognizing unit. This feature is particularly relevant because it

favors the interaction with biological membranes even in the absence of other attractive forces.

The third finding is the relevance of different intermolecular interactions in controlling the binding of AuNPs to biological membranes. H-bonding appears to be very efficient for guanidinium but, according to MD simulations, is not so effective for protonated amines. This is likely because this group's pre-organization is not so optimal for interaction with phosphates. Ion pairing is another interaction that has often been invoked to explain how nanoparticles bind to membranes. However, the outer surface of mammalian cells mostly comprises neutral lipids, and this reduces the significance of this interaction[49]. Indeed, we found that, even for bilayers comprising 10% anionic lipids, ion-pairing is not sufficient to drive nanoparticle binding, as for 4 and 5@AuNPs, and must be accompanied by additional interactions. Remarkably, our results suggest that hydrophobic interactions provided by the alkyl chains of the inner portion of the ligand shell play an important role. This is supported by the fact that, for trimethylammonium AuNPs, effective binding and permeabilization occurred only with long-chain ligands, as with 6 and 7@AuNPs. Indeed, this apparently segregated region of the ligand shell has already been shown to interact with the inner hydrophobic regions of other biomacromolecular entities. The first evidence came from the pioneering studies of Rotello and coworkers[50,51], who demonstrated the ability of the inner alkyl portion of the ligand shell to induce protein denaturation upon AuNP binding. Later on, Alexander-Katz and Stellacci suggested that the "snorkeling" of the alkyl portion of the coating ligands is the key step of the interaction between anionic AuNPs and phospholipid bilayers[5–7,17]. Finally, Stellacci recently reported the ability of anionic AuNPs to induce virus degradation upon binding, thanks to the effect of the ligand shell's inner hydrophobic region[52,53].

In the landscape of nanoparticles for biological applications, ligand-protected noble metal nanoparticles emerge as the only entities capable of featuring precise headgroup functionalization and a flexible inner hydrophobic shell. In addition, the use of "arginine magic" can provide other unique abilities in the interaction with biological membranes. Implications in nanoparticles targeting cells, microorganisms, and viruses may be relevant.

## Methods

**General**. Chemical reagents were purchased from Aldrich at the highest quality and used without further purification. Water was purified using a Milli-Q® and water purification system. Reactions were monitored by TLC developed on 0.25-mm Merck silica gel plates (60 F254) using UV light as visualizing agent and/or heating after spraying ninhydrin. Solvents were of analytical reagent grade, laboratory reagent grade, or HPLC grade. NMR spectra were recorded on a AVIII 500 spectrometer (500 MHz for 1H frequency). ESI-MS was recorded on Agilent Technologies 1100 Series system equipped with a binary pump (G1312A) and MSD SL Trap mass spectrometer (G2445D SL). Synthesis of ligands and nanoparticles are described in the Supplementary Information, Section 1.

The hydrodynamic particle size (dynamic light scattering, DLS) and $Z$-potential were measured with a Malvern Zetasizer Nano-S equipped with a HeNe laser (633 nm) and a Peltier thermostatic system. Measurements were performed at 25°C in water or HEPES 10 mM or HEPES 10 mM, NaCl 100 mM buffer at pH 7.

Thermogravimetric analysis (TGA) was run on 0.4-mg nanoparticle samples using a Q5000 IR instrument from 25 to 1000 °C under a continuous air flow.

Fluorescence spectra and emission recovery experiments were performed in HEPES 10 mM or HEPES 10 mM, NaCl 100 mM buffer at pH 7 on a Varian Cary Eclipse fluorescence spectrophotometer. Both the spectrophotometers were equipped with thermostatted cell holders. Confocal images were taken using a laser scanning confocal microscope (BX51WI-FV300, Olympus) coupled to an Argon laser (IMA-101040ALS, Melles Griot) emitting laser light at 488 nm. The laser beam was scanned on a 512 × 512 px sample area using a 60x water immersion objective (UPLSAPO60xW-Olympus). Fluorescence emission was collected through the same objective, separated from excitation light through a 490-nm long-pass dichroic mirror, and recorded by the PMT with a 510-nm long-pass filter. For fluorescence lifetime experiments, the sample was excited using a frequency-doubled Ti:Sapphire femtosecond laser at 440 nm, 76 MHz (VerdiV5-Mira900-F Coherent), coupled with the BX51WI-FV300 confocal microscope. The emission signal was sent to a single-photon avalanche photodiode (SPAD, MPD, Italy). Before the light enters the photodiode, it passes through a 525/50 band-pass filter. The laser sync and the output of the SPAD were fed to a time-correlated single-photon counting (TCSPC) electronics (PicoHarp 300, PicoQuant, Germany) for the calculation of the emission decay curve. The fitting of the decay curve was performed with the Symphotime software (PicoQuant, Germany), using a two-component exponential model.

Transmission electron microscopy (TEM) was recorded on a FEI Tecnai G12 microscope operating at 100 kV. For cryo-EM, vitrification of samples was performed in liquid ethane cooled at liquid nitrogen temperature using the FEI Vitrobot Mark IV semiautomatic autoplunger. The images were registered with a OSIS Veleta 4 K camera. Bright field cryo-EM was run at −176 °C in a FEI Tecnai G2 F20 transmission electron microscope, working at an acceleration voltage of 200 kV and equipped, relevant for this project, with a field emission gun and automatic cryo-box. The images have been acquired in a low dose modality with a GATAN Ultrascan 1000 2k × 2k CCD.

**Nanoparticle preparation**. Tetraoctylammonium bromide (TOABr, 2.5 eq) was dissolved in toluene and the solution was degassed for 40 min. This solution was used to extract three times an aqueous solution of gold (III) chloride trihydrate (HAuCl₄·3H2O, 1 eq). The combined organic phases were collected in a round-bottom flask along with the remaining solution of TOABr. This mixture was left to stir for about 20 min under an inert atmosphere. Afterward, dioctylamine (DOA, 20 eq) was added all at once. After 1.5 h, the solution was put in an ice bath, then sodium borohydride (NaBH₄, dissolved in milli-Q water, 0.048 mg/μl, 10 eq) was added all at once. After 2 h, the drop of water (which had been used to dissolve sodium borohydride) was removed from the reaction mixture and the desired thiol dissolved in methanol was added. After the formation of the nanoparticles was observed, the mixture was usually stirred for another hour. The nanoparticles were purified by trituration with various organic solvents (each trituration entails the suspension of the nanoparticles in the solvent of choice, sonication, centrifugation, and then removal of the supernatant), and when necessary by gel permeation chromatography. Nanoparticles were characterized by solution ¹H NMR, TEM, DLS, $Z$-potential, and TGA. Details are reported in the Supplementary Information.

**Liposome preparation**. Dichloromethane solutions of phosphatidylcholine (PC) and, when present phosphatidylglycerol (PG), cholesterol (Chol) or nile red, were dried for 4 h under vacuum and then hydrated with a buffered solution of the fluorophore (1 ml, calcein 50 mM, HEPES 10 mM, and NaCl 100 mM, pH 7) under rotation at 42 °C for 40 min. Then, six freeze/thaw cycles were performed, followed

by 15 extrusion filtrations with a polycarbonate membrane (0.1 μm, 19 mm) using an Avanti Polar syringe extruder. Size-extrusion chromatography (G75) with buffer solution (HEPES 10 mM, NaCl 100 mM, pH 7) was used to remove extravesicular fluorophore. The liposome samples were stored at 4 °C.

**Liposome experiments**. Fluorescence recovery experiments were initiated by the addition of a nanoparticle stock solution to a 2-ml buffered solution (HEPES 10 mM, NaCl 100 mM, pH 7.0) containing liposomes (22 μM phospholipid concentration) in a quartz cell. Sample emission at 25 °C was measured followed until no further variations were detected (usually within 5 min). The maximum fluorophore emission was measured after the addition of Triton-X100 in the cell.

**Molecular dynamics simulations**. For this project, we performed molecular dynamics (MD) simulations pertaining five gold nanoparticles (AuNPs) in an aqueous solution and in presence of a PC bilayer. In line with wet-lab experiments, the core size of our AuNPs models was 1.8 nm. The core was functionalized by 60 functionalized thiols. The thiols consisted of a seven-carbon alkyl chain followed by a cationic capping group. The investigated headgroups were: trimethyl ammonium (4@AuNP), dimethyl ammonium (4a@AuNP), monomethyl ammonium (4b@AuNP), ammonium (4c@AuNP), and guanidinium (1@AuNP). The initial geometry of the AuNPs was generated with the NanoModeler webserver[42]. The parameters employed for the internal quasi-static gold atoms were those derived by Heinz, et al.[54], and those used for the gold−sulfur interface were obtained by Pohjolainen, et al.[55] elsewhere. For the coating thiols, the partial charges of all the atoms were calculated with the RESP approach[56,57] by means of the R.E.D. Server[58] whereas the bonded parameters belong to the GAFF force field[59].

We first equilibrated the aforementioned AuNPs in a saline solution. For this, the AuNPs were individually solvated in a box of water (TIP3P parameters[60]) such that the minimum distance between the solute and the edges of the box was 1.0 nm. Then, sodium chloride was added to the system so that the system reached its electroneutrality plus 150 mM ionic strength. We proceeded to minimize the solvent around the AuNPs with the steepest descent method for a maximum of 50,000 steps. Once the system lied in an energetic minimum, it was thermalized to 310 K by a 500-ps-long simulation in the NVT statistical ensemble using the Berendsen thermostat ($\tau_T = 0.1$ ps). Posterior to the thermalization, the system was pressurized to 1 bar with a second 500-ps-long simulation in the NPT ensemble and applying the Berendsen barostat ($\tau_P = 2.0$ ps and $\kappa = 4.5 \times 10^{-5}$ bar⁻¹). Finally, a production run of 100 ns was performed exchanging the Berendsen barostat for its Parrinello−Rahman counterpart[61]. The analysis of the trajectories was done discarding the first 25 ns of simulation for equilibration purposes.

In parallel, we equilibrated the structure of a PC and a PC/PG (9:1) membrane. The starting structure of the membranes was generated with the CHARMM-GUI server and had dimensions of 13 × 13 nm[62]. The phospholipids were then parametrized with the Lipid17 force field[63,64] to ensure compatibility with those employed for the AuNPs in future stages. Each bilayer was then solvated guaranteeing approximately 40 TIP3P water molecules per lipid. Similar to our simulations of AuNPs in saline solutions, these systems were minimized, thermalized, and pressurized. For the production run, we implemented the semi-isotropic Parrinello−Rahman barostat. In this case, the production runs also lasted 100 ns, and the first 25 ns were discarded for equilibration purposes during the analysis.

After having equilibrated the structures of the AuNPs and the membranes, we proceeded to merge them into single systems. In order to do this, the structure of the AuNPs and the membranes were extracted from the last frame of their respective simulations, and we placed the AuNPs 2 nm above the surface of a bilayer. We performed solvation, minimization, thermalization, pressurization, and production runs in accordance with the rest of our simulations. In these cases, the production runs lasted for 1 μs, each.

All our simulations constrained the bonds with the LINCS algorithm[65] and used a timestep of 2 fs. The short-range nonbonded interactions were explicitly calculated for pairs within a distance of 1.2 nm, and the long-range interactions were accounted for with the PME method of fourth-order[66]. Periodic boundary conditions were imposed, and the geometry of the systems was saved every 10 ps. The simulations were performed with Gromacs v2019.2[67,68]. The analysis of the trajectories was done with a mixture of tools already implemented in Gromacs and with a series of in-house scripts using the MDAnalysis Python open library[69]. The number of H-bonds was calculated using the *gmx hbond* utility of Gromacs. The H-bond values labeled as "before binding" were calculated using the 100-ns-long equilibration of 1@AuNP, and the values labeled as "after binding" were calculated using the 300 ns (i.e., 30,000 frames) that followed membrane binding.

## Data availability

The data that support the findings of this study are available from the corresponding authors upon reasonable request. Molecular dynamics trajectories are available at https://github.com/cebasfu93/ArginineMagicGoldNanoparticles.

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

## Acknowledgements
This project received funding from the European Union's Horizon 2020 research and innovation program, under the Marie Skłodowska-Curie grant agreement MMBio (No. 721613). M. D. V. thanks the Italian Association for Cancer Research (AIRC) for financial support (IG "23679"). The authors thank Prof. Camilla Ferrante for granting the access to the laser scanning confocal microscope and a time-correlated single-photon counting (TCSPC) equipment.

## Author contributions
L. M.-B. prepared and characterized the liposomes and performed most of the liposomes experiments; S. F.-U. performed the MD simulations and their analysis. L. M.-B., G. Z., and X. S. prepared the AuNPs. L. M.-B. and I. F. performed the confocal microscopy and TCSPC experiments. R. M. performed the cryoEM experiments. F. M. and M. D. V. designed research and supervised the project. All authors discussed the results and commented on the paper.

## Competing interests
The authors declare no competing interests.
