## [Peer Review File · Communications Chemistry]

Reviewers' comments:

Reviewer #1 (Remarks to the Author):

In this manuscript, the authors reported an experimental and computational investigation of the nondisruptive membrane permeabilization capability of guanidinium-coated AuNPs. The main highlight of this manuscript is the exploitation of the 'arginine magic' penetration process to an AuNP-liposome interaction system. However, the interaction of guanidinium AuNPs did not lead to any membrane penetration but just to an enhanced binding to a lipid bilayer compared to other cationic AuNPs. This leads to an increased membrane permeability which has been detected monitoring the leakage of fluorescent fluorophores from the liposomes. However, the specificity of interaction with PC membranes (which is the most interesting aspect of this manuscript in my opinion) needs further evidences, and more data are required to strengthen these conclusions. In order to claim a specificity of interaction and therefore a specificity of targeting offered by the guanidinium AuNPs, it would be good to investigate their interaction in a two-population liposome mixture encapsulating two different fluorescent probes. If their interaction is selective, only one population will release its cargo. The advantage of using guanidinium AuNPs over other ligand-protected NPs needs to be better demonstrated before acceptance.

In addition, I have also the following comments:

"In this context, properties of small (< 5 nm) gold nanoparticles (AuNPs) coated with a shell of surfactant-like cationic ligands (usually trialkylammonium headgroups) are particularly interesting. These AuNPs were indeed reported to be taken up by cells via an energy-independent (passive) mechanism." The passive uptake on NPs within a membrane trough a non-specific mechanism is never an energy-independent mechanism. Even the cited work (Ref. 19) does not provide a clear mechanism of interaction highlighting the need of further investigations. The authors should better clarify the meaning of their statement.

The main point of this paper is the use of a guanidine coating to investigate the interaction of gold nanoparticles with membranes. However, the introduction is quite vague, and it does not provide a sufficient background information on the arginine effect and its importance in the NPs coating.

The TEM images of AuNPs in figure 1 needs to be taken at higher magnification to properly see the AuNPs and check any possible aggregation events. In particular, in the micrographs number 1 the NPs are not visible. Is there a reason why the AuNPs are more visible in some structures over the other? This puts under question if the synthesis of AuNPs worked better in one way over the others. I also suggest moving all the TEM images in figure 1 to the SI document.

It is not clear how the authors calculated the concentration of ligands in their AuNP samples. Do they assume that all the ligands are attached to the gold surface? Also, depending on the AuNPs average diameter (and therefore surface area), the NPs surface should possess more or less ligand. How has the ligand concentration been calculated?

The authors refer to figure S18 mentioning a ligand/PC concentration ratio, but this ratio is not visible on figure which refers only to calcein release (%) versus concentration of AuNPs' ligand. So, it comes also difficult to see at which concentration the authors identify the level off (plateau) of calcein leakage.

The authors state that the fluorescein release in figure S19 stops after a few minutes. This does not correspond with the kinetic shown in figure as the maximum release is reached after a few seconds. Is there a reason why the authors performed the leakage assay with calcein and the kinetic assay with another fluorophore?

The authors state "Since all the nanoparticles had a similar charge, these results reveal that the interaction between cationic AuNPs and neutral lipid bilayers is governed by factors other than the coating molecules' charge. Namely, the AuNPs' membrane perturbation activity is modulated by variations in the chemical structure, including the nature of the cationic headgroups in equally charged ligands and the features of the underlying chains." It is true that the 1-7 AuNPs formulations have the same charge (positive). However, is the charge density the same for all the formulations? How many coating ligands per surface area? The AuNPs are also different in size. This has an influence on the ligand (and charge) density.

The liposomes permeabilization assays needs to be better discussed by the authors. I also have a few comments on the way how they have been carried out. Buffer and salt mask the surface charges in solution and can have an influence on mediating this interaction. I recommend repeating these experiments in water. Also, the Z potential of both AuNPs and liposomes has been carried out in PBS solution. However, for the interaction investigation, the authors use another type of buffer (HEPES and NaCl). This makes the characterization analysis not relevant. It needs to be repeated with the same solvent in order to provide a better understanding of what is happening during the interaction.

The manuscript is not showing any DLS size distribution profile but just the number of average sizes of the main peak signal. To assess the quality of data and the presence of any other potential size population in the sample, both intensity peaks and correlation function profiles need to be shown.

When discussing the liposomes stability upon nanoparticles addition at different concentration in Supplementary Figure 21, the authors did not observe any changes in the liposome size. However, it is missing a discussion on what is happening at larger AuNPs concentrations. At 1 mM 1@AuNP, the liposome size shift from ~80 nm to ~180 nm. This is not mentioned or discussed in the manuscript.

Figure 3 – confocal microscopy analysis of the interaction. The nano-sized liposomes should be undatable on confocal microscopy and be visible only as undefined background. However, micro-sized aggregates are visible in all samples, control included. This open a question: what are the fluorescent micro-sized aggregates (in green) visible in the images? If they are liposomes, why do liposome form micro-sized aggregates? This is also a good reason to ask for the DLS Intensity signal and correlation function profiles. Also, for the confocal analysis, it would be good to investigate this interaction by using micro-sized liposomes. In this way, it would be possible to visualise and monitor over time the liposome leakage.

Figure 5. The cryo-EM micrographs of PC liposomes only used for the size distribution profile in panel G are not present in the manuscript and they should be included for clarity. Also, The size distribution profile of PC liposomes with 1@AuNPs is not in agreement with Supplementary Figure 21. In the latter figure, there is not a significant change in the liposome diameter which stays unchanged in the region of ~100 nm with a very small error bar (~1nm). However, the size profile in figure 5 tells clearly show a size increase to larger nm (up to at least ~350 nm where the x axis stops). The liposomes are clearly larger than 100 nm in all the three micrographs A-B-C and this trend does

not match with the size profile in panel G.

“The first finding is that the binding of cationic AuNPs to liposomes resulted in their permeabilization in all the cases.” This is not true for 4, 5 and 6@AuNPs formulation as shown from the leakage assay in figure 2.

Reviewer #2 (Remarks to the Author):

See attached.

Reviewer #3 (Remarks to the Author):

In this study, the authors used liposomes as model system to demonstrate that the ability of Guanidium to interact strongly to phospholipid membrane due to their multivalent interaction. It was studied experimentally and by molecular dynamics using gold NPs with different cationic ligands.

This field of research is appealing for many applications related to biomedical research and the use of molecular simulation is undoubtedly getting more and more important to understand mechanisms of interaction between cells and nanoparticles. In this case, the choice of arginine is relevant as numerous NPs showed an enhanced cell uptake when coated with such ligands.

The article is well-written but I found very hard to find the relevant information between the article and the supporting information. Some sections could be more concise (like liposome integrity and resolution of atomically detailed paring between NPS and liposomes) because we lost quickly the important information diluted in the whole manuscript. Many results could be moved to the SI. A big effort needs to be done to improve the quality and the readability of all figures in the ms and I think some of them before the simulation section could be moved to SI.

In the first part related to the NP characterizations, a table gathering important physico-chemical properties is necessary with particle size, hydrodynamic diameter, charge, number of ligand per NPs and density of ligand, organic content). The figure 1 is very confusing with even the formula of the liposomes (PG, PC) without explanation. TEM images are not very useful in the main manuscript and Fig.1C needs to be improved. We are jumping for figure S1 to S19. There are two figures 19. The authors should make an effort to look carefully at these issues.

In the second section “Liposome permeabilization assays, could you please provide more explanation on the determination of ligand/PC and how you find one particle to 30-35 phospholipids. Have you confirmed that Au NPs do not quench the fluorescence of calcein. Also did you check the temperature, incubation time and NP concentration influence on the measurements reported in Fig.2?

In section three “liposome’s structure integrity”, you decided to use liposomes with cholesterol only for 1@AuNP and 4@AuNP (strange acronym by the way). Could you provide at least in the SI the values of Figure 2 for the others AuNP?

In page 6l133, the sentence needs to be rephrased.

I found the experiments done in Fig.3 not very convincing and not very relevant here. Absorbance

and scattering measurements, change of fluorescence lifetime, DLS to confirm no aggregation and degradation of liposomes gathered in one figure (fig.3 and Fig.4) should confirm your conclusion.

The part related to the interaction between AuNP and liposome by cryoTEM was clear and for the next part, again the fig 6 could be improved or removed.

I cannot say anything about MD simulations which is beyond my expertise but I think it will be a good idea to provide a final cartoon gathering all the findings of this study. It is something important in a general journal as Nature communications.

This manuscript by Morillas-Becerril *et al.* tried to address the interaction between liposomes and cationic gold nanoparticles (AuNPs). They reported very interesting results that guanidinium-coated AuNPs can bind to neutral phosphatidylcholine liposomes inducing nondisruptive membrane permeabilization and they also showed that this ability comes from the multivalent H-bond interaction between the phosphate residues and the guanidinium groups using atomistic molecular simulations.

Here are some major points that the authors should address explicitly.

1. Page 6: In the case of 4@AuNP what is the meaning of the negative value for green? What are the effects of the Chol in the case of 2@AuNP and 3@AuNP ? (i.e., what makes a difference comparing to 1@AuNP ?)
2. Page 11: There is a dependence of the liposome diameter (max at 50 ~ 100 nm). What is the physical meaning ? (i.e., what types of factors determine the size ?) With 1@AuNPs the diameter of liposome spreads over 250 nm. Then, what is the role of 1@AuNPs in determining the size of the liposomes ?
3. Page 18, Fig. 9: It is not clear that the figure shows a result for just one 1@AuNP or the average value over all the 1@AuNPs. In the suppl. it says that the binding times are different from each other. How the authors count the number of H-bonds for each 1@AuNP ? Do they have the same pattern of the H-bonding network ?
4. Page 25: It is necessary to simulate 1@AuNPs with PC/PG lipid molecules, and this will strengthen the author's experimental results that PG lipids are not necessary to interact with 1@AuNPs and electrostatic interactions are not the main factor for binding of nanoparticles to membranes.
5. Page 25 and Suppl. Fig. 23: How well the ammonium group or the guanidinium group coordinated with the gold particle ? Please provide the RMSDs of both nanoparticles during the entire simulations.
6. Suppl. movie doesn't show any strong deformation of the lipids bilayer when the nanoparticle binds to it. What would the authors expect that if there are more nanoparticles in the system ? (i.e., what if the concentration of nanoparticles is varied ?). Do the authors expect that 1@AuNPs can translocate through the lipid bilayer ?

There are some minor points which need corrections.

1. Page 11, Fig. 5: White arrows look like yellow arrows in the figure.
2. Page 25: Please check the numbering of nanoparticles, e.g., 4, 4a@AuNP, 4b@AuNP, 4c@AuNP ?
3. Page 26: Please check the numbering of references in the main text. Some of them do not match.
4. Suppl. Fig. 23: Please correct the notation 3@, 3a@ ...

Point to point answers to reviewers comments

Reviewers comments were quoted in red, authors answers are in black

Reviewer 1.

In this manuscript, the authors reported an experimental and computational investigation of the nondisruptive membrane permeabilization capability of guanidinium-coated AuNPs. The main highlight of this manuscript is the exploitation of the 'arginine magic' penetration process to an AuNP-liposome interaction system. However, the interaction of guanidinium AuNPs did not lead to any membrane penetration but just to an enhanced binding to a lipid bilayer compared to other cationic AuNPs. This leads to an increased membrane permeability which has been detected monitoring the leakage of fluorescent fluorophores from the liposomes. However, the specificity of interaction with PC membranes (which is the most interesting aspect of this manuscript in my opinion) needs further evidences, and more data are required to strengthen these conclusions. In order to claim a specificity of interaction and therefore a specificity of targeting offered by the guanidinium AuNPs, it would be good to investigate their interaction in a two-population liposome mixture encapsulating two different fluorescent probes. If their interaction is selective, only one population will release its cargo. The advantage of using guanidinium AuNPs over other ligand-protected NPs needs to be better demonstrated before acceptance.

We understand the reviewer's comment and we realize that the claim of selective targeting we made could be misunderstood. Among cationic nanoparticles, guanidinium ones bind to neural phospholipids while the others do not interact. Hence, strictly speaking, neutral phospholipid-based bilayers are selective for guanidinium nanoparticles, and this is what we demonstrated with our experiments.

On the other hand, guanidinium nanoparticles could be selective for phospholipid-based bilayers in the case that other kinds of neutral bilayers would exist. However, all neutral bilayers made with biological lipids contain the phosphate group. We could not find other commercially available neutral surfactants forming liposomes. In any case, selectivity with respect to these kinds of unnatural systems would be poorly relevant. For these reasons, we could not perform the experiments suggested by the reviewer.

However, to further support the specificity of the interaction between guanidinium nanoparticles and PC we decided to perform the release assays in the presence of sodium dimethylphosphate as potential specific inhibitor (Figure S30). Indeed, we found that substitution of part of NaCl present in the solutions with sodium phosphate (to maintain constant the ionic strength) resulted in a decreased calcein release.

Further evidence of the specificity of the interaction come from the additional computational investigations described later.

We hence added these new results to the paper. In addition, we modified the title and text to better focus the message.

In addition, I have also the following comments:

"In this context, properties of small (< 5 nm) gold nanoparticles (AuNPs) coated with a shell of surfactant-like cationic ligands (usually trialkylammonium headgroups) are particularly interesting. These AuNPs were indeed reported to be taken up by cells via an energy-independent (passive) mechanism." The passive uptake on NPs within a membrane trough a non-specific mechanism is never an energy-independent mechanism. Even the cited work (Ref. 19) does not provide a clear mechanism of interaction highlighting the need of further investigations. The authors should better clarify the meaning of their statement.

In this sentence, we were trying to shortly summarize the literature about the interaction of cationic AuNPs with cells. Of course, we were aware of the fact that energy-independent penetration of nanoparticles in cells is still matter of debate, but we preferred to report entirely the authors claim. We agree however with reviewer 1 that this is not a crucial point since interaction and not internalization is the topic of this paper. Consequently, we removed all the mentions of energy-independent nanoparticles internalization mechanisms from the manuscript.

The main point of this paper is the use of a guanidine coating to investigate the interaction of gold nanoparticles with membranes. However, the introduction is quite vague, and it does not provide a sufficient background information on the arginine effect and its importance in the NPs coating.

We thank the reviewer for highlighting this issue. We revised the introduction by adding a detailed summary of the peculiar properties of guanidinium and their influence on the interaction of polyarginines and poly-guanidinium bearing species with biological membranes.

The TEM images of AuNPs in figure 1 needs to be taken at higher magnification to properly see the AuNPs and check any possible aggregation events. In particular, in the micrographs number 1 the NPs are not visible. Is there a reason why the AuNPs are more visible in some structures over the other? This puts under question if the synthesis of AuNPs worked better in one way over the others.

I also suggest moving all the TEM images in figure 1 to the SI document.

TEM images were removed from figure 1 to SI, were additional TEM pictures were added. The reasons for the different visibility of some NPs is mainly due to slightly different magnifications and to the difficulty in finding the correct focus when small amounts of larger particles are present. Note that NP characterization did not rely only on TEM, but included also DLS, NMR and TGA. All the analyses confirmed that the nanoparticles have a similar structure.

It is not clear how the authors calculated the concentration of ligands in their AuNP samples. Do they assume that all the ligands are attached to the gold surface? Also, depending on the AuNPs average diameter (and therefore surface area), the NPs surface should possess more or less ligand. How has the ligand concentration been calculated?

Reporting AuNP concentrations as ligands concentrations is quite common in the ligand shell-protected gold nanoparticles community, since it gives a clear idea of the amount of "active species" present in the samples. In addition, as stated in the manuscript, the use of the overall ligand concentration allows to normalize differences arising from different nanoparticles average diameter.

The relatively strong sulfur-gold interaction ensures that all the ligands remain attached to the nanoparticles. This is experimentally confirmed by the NMR spectra where the ligand signals are found to be very broad (due to nanoparticle grafting) and the signal of the methylene residue adjacent to the sulfur is so broad to be not visible at all. Diffusion filter NMR spectra retain all the visible ligands signals, confirming that signal broadening is due to nanoparticles grafting. We revised all the nanoparticles characterization section to highlight all this relevant information (Figures S2-S16).

Calculation of the ligand concentrations is performed using the nanoparticles weight concentration and the average chemical formula of the AuNPs. This in turn calculated on the basis of the core size, the atomic density of gold, and the organic content revealed by TGA analyses. We thank the reviewer for noticing that details on concentration calculations were missing and we revised accordingly the SI (section 3).

The authors refer to figure S18 mentioning a ligand/PC concentration ratio, but this ratio is not visible on figure which refers only to calcein release (%) versus concentration of AuNPs' ligand. So, it comes also difficult to see at which concentration the authors identify the level off (plateau) of calcein leakage.

We revised figure S18 (now S20) to clearly identify where we set the levelling off concentration, we also preferred to use the ligand concentration instead of the ligand/PC ratio to identify the samples concentration, and we modified accordingly the discussion.

The authors state that the fluorescein release in figure S19 stops after a few minutes. This does not correspond with the kinetic shown in figure as the maximum release is reached after a few seconds. Is there a reason why the authors performed the leakage assay with calcein and the kinetic assay with another fluorophore?

We thank the reviewer for pinpointing these discrepancies. Indeed, release was in most of the cases complete within one minute: we modified text accordingly.

In all the release experiments we used calcein, which is a fluorescein derivative and the most commonly used dye in release assays. On one occasion, the word "derivative" after "fluorescein" was lost generating the ambiguity reported by reviewer 1, we revised the text removing most of the mentioning of fluorescein.

The authors state "Since all the nanoparticles had a similar charge, these results reveal that the interaction between cationic AuNPs and neutral lipid bilayers is governed by factors other than the coating molecules' charge. Namely, the AuNPs' membrane perturbation activity is modulated by variations in the chemical structure, including the nature of the cationic headgroups in equally charged ligands and the features of the underlying chains." It is true that the 1-7 AuNPs formulations have the same charge (positive). However, is the charge density the same for all the formulations? How many coating ligands per surface area? The AuNPs are also different in size. This has an influence on the ligand (and charge) density.

Reviewer 1 concern is relevant, since the intrinsic differences, albeit small, in average sizes could in principle affect the nanoparticles behavior. Luckily enough, the peculiar structure of the ligand shell protected gold nanoparticles, where thiols are grafted to the surface via the staple motif, grant similar ligand density in all the nanoparticles. We also added, under suggestion of reviewer 3, the new table 1 and table S1 summarizing all the characterization data of the nanoparticles. Inspections of the data confirm that the different behavior observed do not correlate with fluctuations of particles sizes and ligand densities. Similar charge is also confirmed by Z-potential data. In addition, we report the results of an early experiment performed by using two different batches of 1@AuNP with two different average sizes (Figure S22). Results obtained were similar with the two batches, confirming that nanoparticles size is not a relevant parameter.

The liposomes permeabilization assays needs to be better discussed by the authors.

We rewrote the description of the permeabilization assays to make it clearer.

I also have a few comments on the way how they have been carried out. Buffer and salt mask the surface charges in solution and can have an influence on mediating this interaction. I recommend repeating these experiments in water.

We agree with reviewer that salts and buffer might mask the charge interactions. This can be due both to unspecific and specific effects, and indeed we used these specific effects to address in part the reviewer concerns on selectivity, as described earlier.

However, we must point out that permeabilization assay cannot be performed in water. This is because the high (50 mM) concentration of calcein inside the liposomes produces a high ionic strength that must be compensated by adding a salt to the bulk solution during purification, storage and experiments. Otherwise, the resulting osmotic pressure would destroy the liposomes. The conditions we used, including the presence of NaCl in the liposomes solutions, are the ones commonly used for release experiments.

Also, the Z potential of both AuNPs and liposomes has been carried out in PBS solution. However, for the interaction investigation, the authors use another type of buffer (HEPES and NaCl). This makes the characterization analysis not relevant. It needs to be repeated with the same solvent in order to provide a better understanding of what is happening during the interaction.

Reviewer is correct, even if we must note that NaCl 100 mM is the main component of both the buffer systems used (PBS and HEPES/NaCl). This was the reason why we considered the characterizations performed in PBS relevant in HEPES/NaCl. At any rate, to remove any concern, we repeated Z-potential measurements for liposomes and nanoparticles in the same buffer of the interaction experiments (Figure S23).

The manuscript is not showing any DLS size distribution profile but just the number of average sizes of the main peak signal. To assess the quality of data and the presence of any other potential size population in the sample, both intensity peaks and correlation function profiles need to be shown.

Required data were added to figure 3 and to the SI, Figures S22 and S24.

When discussing the liposomes stability upon nanoparticles addition at different concentration in Supplementary Figure 21, the authors did not observe any changes in the liposome size. However, it is missing a discussion on what is happening at larger AuNPs concentrations. At 1 mM 1@AuNP, the liposome size shift from ~80 nm to ~180 nm. This is not mentioned or discussed in the manuscript.

We did not discuss those data because the observed size increase occurred at such a high nanoparticles concentration that unbalance of osmotic pressure, with consequent liposomes degradation, was possible. We revised the manuscript and added a comment to this observation. In addition, we repeated the experiments with 4@AuNP (Figure S25): the result show that with these not interacting NPs the liposomes size remains unaltered even at the highest concentration.

Figure 3 – confocal microscopy analysis of the interaction. The nano-sized liposomes should be undatable on confocal microscopy and be visible only as undefined background. However, micro-sized aggregates are visible in all samples, control included. This open a question: what are the fluorescent micro-sized aggregates (in green) visible in the images? If they are liposomes, why do liposome form micro-sized aggregates? This is also a good reason to ask for the DLS Intensity signal and correlation function profiles. Also, for the confocal analysis, it would be good to investigate this interaction by using microsized liposomes. In this way, it would be possible to visualise and monitor over time the liposome leakage.

Indeed, resolution of confocal microscopy is not enough to allow the detection of liposomes. Still, if the solution is diluted enough and with the proper setup it is possible to detect the light emitted by a single liposome, and the apparent size will correspond to one or two pixels

of the image. This is what is mainly seen in figure 3. If several liposomes are close, even if not aggregated, enlightened pixels might appear as a micro-sized entity. However, liposomes aggregation is excluded by DLS experiments performed in the same conditions.

We modified discussion of figure 3 to make these points clearer, and we also moved figure 3, and the relative discussion, to SI as suggested by reviewer 3 (Figure S28).

Figure 5. The cryo-EM micrographs of PC liposomes only used for the size distribution profile in panel G are not present in the manuscript and they should be included for clarity. Also, The size distribution profile of PC liposomes with 1@AuNPs is not in agreement with Supplementary Figure 21. In the latter figure, there is not a significant change in the liposome diameter which stays unchanged in the region of ~100 nm with a very small error bar (~1nm). However, the size profile in figure 5 tells clearly show a size increase to larger nm (up to at least ~350 nm where the x axis stops). The liposomes are clearly larger than 100 nm in all the three micrographs A-B-C and this trend does not match with the size profile in panel G.

We reported the cryoEM micrographs requested in figure S19. The discrepancy of cryoEM and DLS data arises from the different nature of the two investigations. CryoEM is performed on frozen samples, and this may partially affect the samples, while DLS is performed in solution in the same conditions of the calcein release experiments.

“The first finding is that the binding of cationic AuNPs to liposomes resulted in their permeabilization in all the cases.” This is not true for 4, 5 and 6@AuNPs formulation as shown from the leakage assay in figure 2.

We did not mean that all AuNPs permeabilize liposomes, but that permeabilization is the consequence of binding. We rephrased the sentence to make it clearer.

Reviewer 2.

This manuscript by Morillas-Becerril et al. tried to address the interaction between liposomes and cationic gold nanoparticles(AuNPs). They reported very interesting results that guanidinium-coated AuNPs can bind to neutral phosphatidylcholine liposomes inducing nondisruptive membrane permeabilization and they also showed that this ability comes from the multivalent H-bond interaction between the phosphate residues and the guanidinium groups using atomistic molecular simulations.

Here are some major points that the authors should address explicitly.

1. Page 6: In the case of 4@AuNP what is the meaning of the negative value for green?

In the case of 4@AuNP with PC/Chol liposomes we observed a very small decrease of emission in the release experiments. However, taking into consideration the errors this negative value is not significant.

What are the effects of the Chol in the case of 2@AuNP and 3@AuNP ? (i.e., what makes a difference comparing to 1@AuNP ?)

We added to figure 2 the results of the experiments performed with PC/Chol with all the nanoparticles. Taking errors into account, there is no relevant difference in the calcein release from PC or PC/Chol liposomes.

2. Page 11: There is a dependence of the liposome diameter (max at 50 ~ 100 nm). What is the physical meaning ? (i.e., what types of factors determine the size ?) With 1@AuNPs the diameter of liposome spreads over 250 nm. Then, what is the role of 1@AuNPs in determining the size of the liposomes ?

As described in the caption, Figure 5g reports the size distribution of PC liposomes alone and in the presence of 1@AuNP. The presence of liposomes of different sizes distributed around a most populated one (79 nm, see the text) is typical of these preparations. The goal of the picture is to demonstrate that addition of AuNP does not produce substantial changes of the size distributions except for the formation of a small fraction of larger liposomes, hence liposomes maintain essentially their structure. In addition, liposomes integrity is confirmed by DLS, emission lifetimes and also by confocal microscopy.

3. Page 18, Fig. 9: It is not clear that the figure shows a result for just one 1@AuNP or the average value over all the 1@AuNPs. In the suppl. it says that the binding times are different from each other. How the authors count the number of H-bonds for each 1@AuNP ? Do they have the same pattern of the H-bonding network ?

In the original manuscript, Figure 9 discussed the binding dynamics and H-bond network for one of the four replica simulations of 1@AuNP interacting with a pure POPC bilayer. Now, we have analyzed the other three replica simulations and added them to the revised version of the manuscript. Figures S33, S34, and S35 show that the binding process of the other three replicas is comparable in terms of the H-bond networks established and the structural distortions induced on the monolayer and the membrane. We have edited the Results section and the Supplementary Information to distinguish more clearly the results from each replica.

Moreover, we have added Table S4 summarizing the number of H-bonds formed between the coating thiols and the POPC lipids before and after the nanoparticle-membrane binding event. There, we report the number of H-bonds formed between i) the 60 ϵ nitrogen atoms in 1@AuNP and the oxygen atoms in the solvent, ii) the 60 ϵ nitrogen atoms in 1@AuNP and the phosphate group in the lipids, iii) the 120 η nitrogen atoms in 1@AuNP and the oxygen atoms in the solvent, and iv) the 120 η nitrogen atoms in 1@AuNP and the phosphate group in the lipids. The number of H-bonds was calculated using the *gmx hbond* utility implemented in Gromacs. The values labeled as “before binding” were calculated using the 100 ns-long equilibrations of 1@AuNP, and the values labeled as “after binding” were calculated using the 300 ns (i.e. 30000 frames) that followed membrane binding. This procedure is now explained in the Methods section.

4. Page 25: It is necessary to simulate 1@AuNPs with PC/PG lipid molecules, and this will strengthen the author's experimental results that PG lipids are not necessary to interact with 1@AuNPs and electrostatic interactions are not the main factor for binding of nanoparticles to membranes.

We thank the reviewer for this observation. The revised version of the manuscript includes simulations of a POPC:POPG bilayer with a lipid molar ratio of 9:1 to match the experimental conditions. Specifically, we have performed four new 1 μ s-long replica simulations of 1@AuNP interacting with the latter POPC:POPG membrane. As expected by the reviewer, these simulations showed 1@AuNP binding spontaneously (and permanently) to the membrane, corroborating that electrostatic interactions are not the main factor for the binding of these nanoparticles to membranes. We have analyzed these new simulations in the same way as those using pure POPC bilayers, and we have the results to the manuscript.

We have also modified the Methods section to report the set up of the simulations. Lastly, we appended Figures S43-S48 and Table S5 with data on each of the POPC:POPG replica simulations. These figures report the distance between the AuNPs and the membranes over time, the H-bond networks formed, the reorganization of the monolayer and the membrane, the changes in membrane thickness, and the lipid order parameters.

5. Page 25 and Suppl. Fig. 23: How well the ammonium group or the guanidinium group coordinated with the gold particle ? Please provide the RMSDs of both nanoparticles during the entire simulations.

In accordance with the reviewer's comment, we have calculated the RMSD of the nanoparticles during the entire simulations and included it as Figure S32. It is important to note, however, that classical molecular dynamics (MD) simulations, by definition, do not allow the cleavage or formation of chemical bonds. That is, the connectivity of the molecules is predefined by how the system is parametrized. Here, we have used models of AuNPs in which the grafted thiols are covalently bound to gold atoms on the surface of the metallic core. These models offer an effective strategy for studying structural and dynamical features of monolayer-protected AuNPs as they are based on crystallographic data (see e.g. Franco-Ulloa, et al. *J. Chem. Theory Comput.* DOI: <https://pubs.acs.org/doi/abs/10.1021/acs.jctc.8b01304> and Riccardi, L., et al. *Chem Cell Press*, DOI: 10.1016/j.chempr.2017.05.016). With this into consideration, the reported RMSD reflects the flexibility and packaging of the coating monolayers. Indeed, Figures S23 and S25 show that all our model systems retain their structural integrity throughout the simulations.

6. Suppl. movie doesn't show any strong deformation of the lipids bilayer when the nanoparticle binds to it. What would the authors expect that if there are more nanoparticles in the system ? (i.e., what if the concentration of nanoparticles is varied ?). Do the authors expect that 1 @AuNPs can translocate through the lipid bilayer ?

The reviewer is right at pointing out that there were not any strong deformations on the membrane upon nanoparticle binding. This observation is corroborated by a nearly constant membrane thickness (Figures S37 and S38) and lipid order parameter (Figure S39 and S40). However, this landscape could change in the presence of multiple guanidinium-bearing nanoparticles. Despite their positive charge, multiple AuNPs could aggregate on the surface of the bilayer due to the amphiphilic character of guanidinium ions (see e.g. Sakai, N., et al. *J. Am. Chem. Soc.* DOI: 10.1021/ja037601l). The aggregation of AuNPs on the membrane's surface could subsequently cause the gluing of neighboring liposomes as evidenced in our cryoEM images. Else, the AuNPs could also cooperate to foment their translocation as evidenced in arginine-rich cell-penetrating peptides (see e.g. Yesylevskyy, S., et al. *Biophys. J.* DOI: 10.1016/j.bpj.2009.03.059). In fact, our simulations show a reorientation of the phosphatidylcholine headgroups (Figures 8 and S41) that places the anionic phosphate groups closer to the cationic nanoparticle. This subtle distortion causes a local polarization of the bilayer, potentially affecting the adsorption and translocation rates of the AuNPs with the membrane.

There are some minor points which need corrections.

1. Page 11, Fig. 5: White arrows look like yellow arrows in the figure.

Corrected

2. Page 25: Please check the numbering of nanoparticles, e.g., 4, 4a@AuNP, 4b@AuNP, 4c@AuNP?

We have corrected and verified the numbering of all the nanoparticles throughout the manuscript. 4@AuNP, 4a@AuNP, 4b@AuNP, and 4c@AuNP refer to gold nanoparticles

coated with trimethylammonium, dimethylammonium, monomethylammonium, and ammonium, respectively.

3. Page 26: Please check the numbering of references in the main text. Some of them do not match.

We revised the text and corrected the reference numbering.

4. Suppl. Fig. 23: Please correct the notation 3@, 3a@ ...

We have corrected the labels on Figure S23 accordingly.

Reviewer 3.

In this study, the authors used liposomes as model system to demonstrate that the ability of Guanidium to interact strongly to phospholipid membrane due to their multivalent interaction. It was studied experimentally and by molecular dynamics using gold NPs with different cationic ligands.

This field of research is appealing for many applications related to biomedical research and the use of molecular simulation is undoubtedly getting more and more important to understand mechanisms of interaction between cells and nanoparticles. In this case, the choice of arginine is relevant as numerous NPs showed an enhanced cell uptake when coated with such ligands.

The article is well-written but I found very hard to find the relevant information between the article and the supporting information. Some sections could be more concise (like liposome integrity and resolution of atomically detailed pairing between NPS and liposomes) because we lost quickly the important information diluted in the whole manuscript. Many results could be moved to the SI. A big effort needs to be done to improve the quality and the readability of all figures in the ms and I think some of them before the simulation section could be moved to SI.

In the first part related to the NP characterizations, a table gathering important physico-chemical properties is necessary with particle size, hydrodynamic diameter, charge, number of ligand per NPs and density of ligand, organic content). The figure 1 is very confusing with even the formula of the liposomes (PG, PC) without explanation. TEM images are not very useful in the main manuscript and Fig.1C needs to be improved. We are jumping for figure S1 to S19. There are two figures 19. The authors should make an effort to look carefully at these issues.

We thank reviewer 3 for suggestions. We removed TEM images from figure 1, as also suggested by reviewer 1, and added a table (Table 1 and table S1) with nanoparticles chemico-physical properties. We also revised all the SI to improve layout and easiness to find information.

In the second section "Liposome permeabilization assays, could you please provide more explanation on the determination of ligand/PC and how you find one particle to 30-35 phospholipids.

We realized that this parameter is quite meaningless as it depends on the particle size, and we preferred to use only the ligand/PC ratio. Text and SI were revised to better explain our calculation, as described in the answer to the comments of reviewer 1.

Have you confirmed that Au NPs do not quench the fluorescence of calcein. Also did you check the temperature, incubation time and NP concentration influence on the measurements reported in Fig.2?

Effect of incubation time and NP concentration were investigated were reported in Figures S20 and S21 (former S18 and S19). We have added now the results obtained at different temperature (Figure S31), apparently increasing the temperature levels off the effects produced by all the nanoparticles. We believe that this is due to the fact that thermal agitation can enhance the relevance of the hydrophobic interactions but more investigation will be necessary to understand this point.

In section three "liposome's structure integrity", you decided to use liposomes with cholesterol only for 1@AuNP and 4@AuNP (strange acronym by the way). Could you provide at least in the SI the values of Figure 2 for the others AuNP?

We provided the values for all the AuNPs, now added to Figure 2, remarkably, no relevant effects were observed also with the other NPs. See response to comments of referee 2. The symbol @ is used by the nanoparticles community to indicate "inclusion". 1@AuNP hence means a gold core included by a 1 shell.

In page 61133, the sentence needs to be rephrased.

Done.

I found the experiments done in Fig.3 not very convincing and not very relevant here. Absorbance and scattering measurements, change of fluorescence lifetime, DLS to confirm no aggregation and degradation of liposomes gathered in one figure (fig.3 and Fig.4) should confirm your conclusion.

We followed this suggestion by removing figure 3 and relative discussion from the main text (Now figure S28) and preparing a new figure 4 (now figure 3) including all the required experiments.

The part related to the interaction between AuNP and liposome by cryoTEM was clear and for the next part, again the fig 6 could be improved or removed.

We moved also figure 6 to the SI, now it is figure S29.

I cannot say anything about MD simulations which is beyond my expertise but I think it will be a good idea to provide a final cartoon gathering all the findings of this study. It is something important in a general journal as Nature communications.

Reviewers' comments:

Reviewer #1 (Remarks to the Author):

The authors have improved the revised version with more data and discussion. My questions have been addressed and the manuscript can be accepted for publication.

Only a few minor remark:

- The discrepancy of cryoEM and DLS data is still unclear. The authors ascribe this to the different nature of the two techniques specifying that "the cryoEM is performed on frozen samples, and this may partially affect the samples".

CryoEM is performed by verifying the sample which technique highly preserve the vesicles structures as is also shown in the micrographs. So, the sample damage should be minimal. Also, it is unclear how sample damage due to verification, can make the vesicles larger.

- The calcein leakage assay can be performed in water and in absence of NaCl by compensating the osmotic pressure with a sucrose solution.

Reviewer #2 (Remarks to the Author):

The revision is well written and the authors have clarified several issues mentioned in the referee's comments. The manuscript becomes more readable, and I recommend publication.

Reviewer #3 (Remarks to the Author):

The authors made a substantial efforts to reply to the comments for the three reviewers and improved a lot the readability. The experimental data are well supported by MD simulations. I consider this work could be published as it is in Communication chemistry

Point to point answers to reviewers comments

Reviewers comments were quoted in red, authors answers are in black

Reviewer 1.

The discrepancy of cryoEM and DLS data is still unclear. The authors ascribe this to the different nature of the two techniques specifying that "the cryoEM is performed on frozen samples, and this may partially affect the samples". CryoEM is performed by verifying the sample which technique highly preserve the vesicles structures as is also shown in the micrographs. So, the sample damage should be minimal. Also, it is unclear how sample damage due to verification, can make the vesicles larger.

The reviewer is right. Since we were not able to detect the presence of larger liposomes with DLS and absorbance measurements, we concluded that those observed in cryEM should have been artifacts arising from sample preparation. Reconsidering the results on the basis of his comments, we concluded that the larger liposomes could be the result of a fusogenic activity of the guanidinium nanoparticles. We modified the description of the cryoEM experiments accordingly.

The calcein leakage assay can be performed in water and in absence of NaCl by compensating the osmotic pressure with a sucrose solution.

Reviewer 1 is right again. We apologize for not having considered such a simple solution. We prepared liposomes without NaCl using glucose to compensate the osmotic pressure and repeated the calcein release experiment. Results are reported in new figure S23 and show that the absence of NaCl does not affect sensibly the nanoparticles' activity.

REVIEWERS' COMMENTS:

Reviewer #1 (Remarks to the Author):

The authors addressed all the comments and I recommend the manuscript for publication.